# Global and regional importance of the direct dust-climate feedback

Jasper F. Kok [1], Daniel S. Ward [2], Natalie M. Mahowald [3] & Amato T. Evan [4]

Feedbacks between the global dust cycle and the climate system might have amplified past climate changes. Yet, it remains unclear what role the dust–climate feedback will play in future anthropogenic climate change. Here, we estimate the direct dust–climate feedback, arising from changes in the dust direct radiative effect (DRE), using a simple theoretical framework that combines constraints on the dust DRE with a series of climate model results. We find that the direct dust–climate feedback is likely in the range of $-0.04$ to $+0.02$ Wm$^{-2}$ K$^{-1}$, such that it could account for a substantial fraction of the total aerosol feedbacks in the climate system. On a regional scale, the direct dust–climate feedback is enhanced by approximately an order of magnitude close to major source regions. This suggests that it could play an important role in shaping the future climates of Northern Africa, the Sahel, the Mediterranean region, the Middle East, and Central Asia.

[1] Department of Atmospheric and Oceanic Sciences, University of California, Los Angeles, CA 90095, USA. [2] Program in Atmospheric and Oceanic Sciences, Princeton University, Princeton, NJ 08544, USA. [3] Department of Earth and Atmospheric Sciences, Cornell University, Ithaca, NY 14850, USA. [4] Scripps Institution of Oceanography, University of California, La Jolla, San Diego, CA 92037, USA. Correspondence and requests for materials should be addressed to J.F.K. (email: jfkok@ucla.edu)

Mineral dust is likely the most abundant aerosol type by mass in the atmosphere[1]. It affects the climate system in numerous ways, including by scattering and absorbing radiation, serving as a nuclei for cloud formation, and fertilizing ecosystems upon deposition[2–4]. Consequently, changes in the atmospheric dust loading have been hypothesized to produce a substantial radiative forcing of the climate system[5–7]. The global dust cycle is itself also highly sensitive to changes in climate[8,9], as evidenced by global dust deposition being several times larger during glacial maxima than during interglacials[10], and by the large variability of the global dust cycle over the observational record of the past 50 years[6,11,12].

This sensitivity to past climate changes raises the question of how the global dust cycle will respond to projected future climate changes, and of whether the resulting dust–climate feedback will oppose or enhance those climate changes. However, past assessments of the dust cycle response to future climate changes have yielded divergent results, and there is no consensus on whether the dust–climate feedback will enhance or oppose future climate changes[8,13–15]. Indeed, an accurate quantification of the dust–climate feedback is hindered by several uncertainties, including in future changes to wind and precipitation patterns[11,16], the effect of $CO_2$ fertilization on desert extent[14,15], the dust radiative effect in the present climate[1], and the empirical nature of dust emission parameterizations in global circulation models[17]. These parameterizations generally use so-called preferential dust source functions that tune emissions to better reproduce the present climate's dust cycle[18–20]. Since these source functions are invariant to climate, whereas the geological record shows that source regions are quite sensitive to climate[6,10], using a source function in simulations of the dust cycle's response to future climate change could produce large systematic errors[21].

Although interactions between climate and the global dust cycle remain highly uncertain, recent advances now permit an order-of-magnitude estimation of the direct dust–climate feedback. In particular, a physically based dust emission scheme can now reproduce the global dust cycle without the use of an empirical source function[17,21]. Furthermore, the coupling of the nitrogen and carbon cycles in climate models[22] has enabled nitrogen limitation to more realistically constrain the effects of $CO_2$ fertilization on desert extent[23]. Finally, recent work has constrained the dust direct radiative effect (DRE) in the present climate[1].

Here, we leverage these advances in order to estimate the direct dust–climate feedback. We do so by using a simple theoretical framework that combines constraints on the dust DRE[1] with model simulations of how that DRE is distributed regionally, and with a series of climate model results on the response of the change in global dust loading in response to future climate change. A subset of these simulations use both coupled carbon–nitrogen cycles[22,24] and the more physically based dust-emission parameterization that does not require a source function[17,21]. We find that the global direct dust–climate feedback is of the order of a percent of the total physical feedbacks in the climate system, and that it is approximately an order of magnitude larger close to source regions, where it could play an important role in shaping future climate.

## Results

**Theory of the dust–climate feedback.** We first develop a theoretical framework for estimating the dust–climate feedback from climate model simulations. Feedbacks in the climate system are defined according to the equation:[25,26]

$$\Delta R = \Delta F + \lambda \Delta T, \qquad (1)$$

where $\Delta R$ (Wm$^{-2}$) is the radiative imbalance at top-of-atmosphere (TOA), $\Delta F$ (Wm$^{-2}$) is the (instantaneous) radiative forcing disequilibrating the Earth's radiation budget (such as the positive forcing due to increasing greenhouse gas concentrations), and $\Delta T$ (K) is the change in the globally averaged surface temperature. The climate feedback parameter $\lambda$ (Wm$^{-2}$ K$^{-1}$) is then defined as[25,26]

$$\lambda = \frac{\partial R}{\partial T}. \qquad (2)$$

For the dust–climate feedback, this change in the radiative forcing $(\partial R)$ corresponds to the change in dust radiative effects that occurs per unit change in the globally averaged surface temperature $(\partial T)$. Dust affects Earth's radiative balance directly through interactions with radiation, termed the direct radiative effect (DRE), and indirectly through interactions with clouds, biogeochemistry, and the cryosphere[2,3]. These indirect effects are poorly understood, with large and poorly quantified uncertainties[2,3]. We therefore restrict our discussion to the direct dust–climate feedback arising from changes in the dust direct radiative effect only, which is better understood and has a better quantified uncertainty[1].

A change in global dust loading changes the dust DRE, which constitutes a radiative forcing. Since the dust DRE is approximately linear in the global dust loading[1,6], the dust direct radiative forcing at TOA relative to the present-day climate, $\Delta\zeta$, can be approximated as:

$$R_{\text{dust}}(t) = \Delta\zeta(t) \approx \zeta_0 \frac{L(t) - L_0}{L_0}, \qquad (3)$$

where $L(t)$ is the global dust loading at a future time $t$, and $L_0$ is the present-day dust loading. Furthermore, $\zeta_0$ is the dust DRE in the present-day climate, which was recently constrained to a median value of $-0.20$ Wm$^{-2}$ with a 95% confidence interval (CI) of $-0.48$ to $+0.20$ Wm$^{[-2}$ [1]. Combining Eqs. (2) and (3) then yields the global direct dust–climate feedback as

$$\lambda_{\text{dust}} \equiv \frac{\partial R_{\text{dust}}(t)}{\partial T(t)} \cong \zeta_0 \frac{\Delta L(t)/L_0}{\Delta T(t)} \equiv \zeta_0 \kappa, \qquad (4)$$

where we defined $\kappa \equiv \frac{\Delta L/L_0}{\Delta T}$ as the fractional change in the global dust loading per degree global surface temperature change.

The direct dust–climate feedback is likely to be substantially larger near source regions than it is on a global basis. We therefore also quantify the regional direct dust–climate feedback $(\tilde{\lambda}_{\text{dust}})$ as the regional change in the dust DRE $(\Delta\tilde{\zeta})$ per unit globally averaged surface temperature change $(\Delta T)$, where the tilde denotes a regional value at longitude $\theta$ and latitude $\phi$. Using Eq. (4), this can be written as

$$\tilde{\lambda}_{\text{dust}}(\theta, \phi) = \tilde{\zeta}_0(\theta, \phi)\tilde{\kappa}(\theta, \phi). \qquad (5)$$

Note that some previous studies have defined the regional feedback relative to the local temperature change[26]. Since aerosols are largely advected from upwind locations, and since aerosol feedbacks do not arise from a local thermodynamic response, there is less causal relation between temperature and aerosol loading at a given location. Consequently, defining the regional feedback relative to the local temperature change would not capture the physical processes (e.g., upwind changes in wind speed, vegetation, soil moisture, and precipitation) that ultimately drive local changes in aerosol concentration and radiative effects. We have therefore defined the regional direct dust–climate feedback as relative to the change in global surface temperature.

Equation (5) should thus be interpreted as the change in the regional energy balance induced by the response of dust aerosol loading to global climate changes, for which changes in the globally averaged surface temperature serves as a proxy.

Considering the divergence between simulations of the global dust cycle response to climate changes[5,13,14,27], and the inability of most models to reproduce historical changes in regional dust loading[27], the skill of current models in predicting future changes in dust content at a specific location can be considered very limited. Therefore, we simply take the local change in dust loading per degree globally averaged surface temperature change as equal to its global value (i.e., $\tilde{\kappa}(\theta,\phi) \cong \kappa$). The spatial variability in the calculated $\tilde{\lambda}_{dust}$ (Eq.(5)) will thus be driven by variability in $\tilde{\zeta}_0(\theta,\phi)$, the regional dust DRE in the present climate, which we determine by combining constraints on the global dust DRE[1] with an ensemble of four climate model simulations of how that DRE is distributed regionally (see Methods). The regional enhancement of the direct dust–climate feedback over its globally averaged value is then,

$$\tilde{E}_\lambda(\theta,\phi) = \tilde{\zeta}_0(\theta,\phi)/\zeta_0. \qquad (6)$$

**Climate model simulations.** In order to estimate the direct dust–climate feedback using the framework above, we need to quantify the change in dust loading per degree globally averaged surface temperature change ($\kappa$). We do so using an ensemble of the subset of simulations from the Coupled Model Intercomparison Project Phase 5 (CMIP5) that included prognostic treatments of dust aerosols[28,29] (see Methods). However, these simulations have three important limitations. First, CMIP5 simulations include projected future land use changes, such that it is not clear which part of the simulated dust loading change is due to land use changes, which are considered anthropogenic emissions and thus a radiative forcing[30], and which part is due to the response of the dust cycle to climate changes, which gives rise to the dust–climate feedback. Second, the ability of many CMIP5 models to project future dust cycle changes is limited by their use of empirical dust emission parameterizations to parameterize the spatial variability of dust emissions[21]. Third, although many CMIP5 models are able to reproduce important features of the historical warming trends[31], CMIP5 models have not been shown able to reproduce historical trends in dust loading without forcing changes in dust sources[6,27], such that the usefulness of CMIP5 simulations in forecasting the dust cycle's response to future climate changes is uncertain[11].

To address these limitations of CMIP5 simulations in estimating $\kappa$, we perform additional simulations with the Community Earth System Model (CESM). For these simulations, we (i) set future land use equal to that of the present day to isolate the dust cycle's response to climate change[21], (ii) use a recent physically based dust emission scheme that does not require a source function[17], and (iii) evaluate whether the model can reproduce past changes in dust emissions. We describe these CESM simulations in detail in the Methods section, and summarize them here briefly. We conducted three simulations with both the default (BASE) CESM dust emission module[19] and the more physically based (PHYS) dust emission model of Kok et al.[17], for a total of six simulations. This new emission scheme, hereafter referred to as K14, has been shown to both better reproduce small-scale dust emission measurements[17] and to better reproduce dust AOD retrievals when implemented in CESM (see ref. [21] and Supplementary Fig. 1). Two of the three simulations with each dust module were driven by reanalysis meteorology, and were used to test the model's ability to reproduce historical changes in dust emissions, as captured by

two data sets that serve as a proxy for dust emission. The first data set is a satellite-derived long-term record of dust aerosol optical depth (AOD) off the West African coast around Cape Verde. Dust AOD in this region is dominated by the main dust plume blowing across the Atlantic Ocean from North Africa, and is thus a proxy for the intensity of North African dust emissions[32]. Our second data set consists of measurements of AOD from the AERONET network of ground-based sun photometers[33], namely at eight stations near dust source regions for which long–term changes in AOD are due to changes in dustiness (see Methods). The third simulation used the land component of CESM, the Community Land Model (CLM)[24], to simulate the response of the global dust emission rate to future climate changes, which we use as a proxy for global dust loading[6], and combine with Eqs. (4) and (5) to estimate the direct dust–climate feedback (see Methods).

We found that CESM's BASE dust module substantially underestimates both the AOD off the West-African coast, and its relative rate of decline over the past few decades (Fig. 1a), although this is somewhat sensitive to which reanalysis meteorology data set is used[34]. Using the PHYS dust module with the physically based K14 emission scheme helps to resolve both these problems. First, the magnitude of the dust AOD is brought in better agreement with measurements, because the K14 scheme shifts emissions to the more erodible regions in Western Africa (Figs. 3, 4 in ref. [21]), which results in a greater dust AOD over the tropical North Atlantic. Second, CESM with the PHYS module better captures the historical decline in dust AOD since the 1980s, which is likely largely due to a historical decrease in wind speed over the North African source regions[11] (Supplementary Fig. 3). CESM's improved performance with the PHYS module is thus likely due in part to a more accurate representation of Western Africa source regions, as well as possibly due to improvements in the scaling of dust emissions with wind speed, and the increased sensitivity of dust emissions to climate changes (particularly due to changes in soil moisture) in K14.

The comparison of CESM simulations against AERONET measurements shows that the BASE dust module has limited skill in reproducing observed long-term trends (Supplementary Fig. 1), producing a correlation coefficient of 0.50 between measured and modeled long-term trends (Fig. 1b). Using the PHYS dust module improves the simulated long-term trend at most stations, resulting in an improved correlation coefficient of 0.83.

These results suggest that CESM's PHYS dust module produces improved skill in capturing observed long-term changes in dust emissions, when forced by reanalysis meteorology, which provides some limited confidence in simulations of future changes in the global dust emission rate (see Methods for remaining important limitations on the model simulations). The future simulations forecast an increase in the global dust emission rate (Supplementary Fig. 4), which is driven by two factors. The first factor is a slight increase in wind speed over most North African dust source regions, which is within the range of CMIP5 simulation results (Fig. 3b in ref. [11]), although the mean of CMIP5 simulations predicts a slight decrease in dust emissions due to North African wind speed changes. The second factor is a decrease in soil moisture in almost all source regions[16,35] (Supplementary Fig. 5), and particularly at the margins of the major African and Asian source regions. Furthermore, the enhanced sensitivity of the K14 dust emission parameterization to the soil state causes regions with substantial changes in soil moisture to show larger differences in the response of the dust flux between the old and the new parameterizations. Consequently, CESM with the PHYS emission module predicts an enhanced sensitivity of the dust cycle to future climate changes (Fig. 2). That is, we find that the increase in the global dust

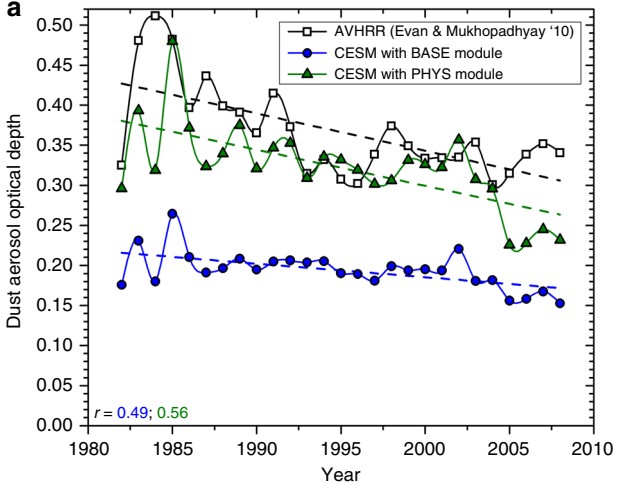

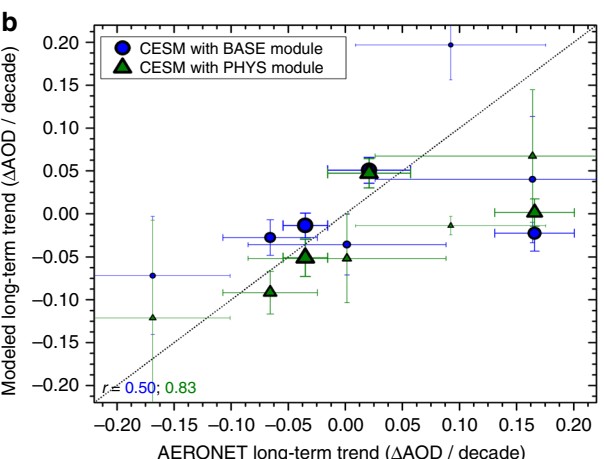

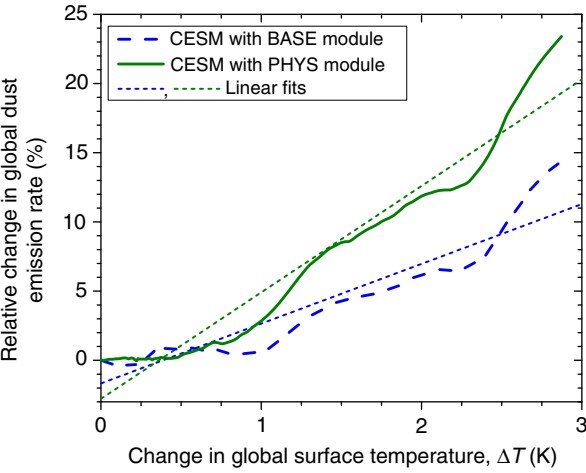

**Fig. 2** CESM simulations of the response of the global dust emission rate to future climate change. Shown are changes in the 25-year running average of the global dust emission rate relative to the period 1976–2000, as a function of global surface temperature change $\Delta T$, simulated using CLM with the BASE and PHYS dust modules. The dashed lines denote linear least-squares fits. The additional dust emission physics in the PHYS module almost doubles the dust cycle's response to climate change

**Fig. 1** Comparison of CESM simulations against historical long-term changes in dust aerosol optical depth. **a** Shown is the AVHRR satellite record of annually averaged dust AOD (black line and squares) over the dust-dominated region of 10–20°N and 20–30°W, off the West-African coast[27,32]. The CESM simulation with the BASE dust emission module (blue line and circles) poorly reproduces the magnitude and historical trend of the dust AOD, whereas CESM with the PHYS emission module (green line and triangles) better reproduces both the magnitude of the dust AOD and its long-term trend. The slope of the least-square linear fit lines (dashed lines) are −0.047 ± 0.011 per decade (corresponding to −13 ± 3% per decade) for the AVHRR data, −0.017 ± 0.005 per decade (corresponding to −9 ±2% per decade) for the BASE-AVHRR simulation, and −0.045 ± 0.010 per decade (corresponding to −14 ±3% per decade) for the PHYS-AVHRR simulation. **b** Shown are the modeled long-term trends in AOD with the BASE and PHYS emission modules, compared against the measured trends at eight AERONET stations (see Supplementary Figs. 1 and 2 for results from the individual stations) for which these long-term changes are dominated by changes in dust aerosols (see Methods). The size of the symbols is proportional to the number of years of data for each station, and error bars were obtained from least-squares fitting of the modeled annually averaged AOD at each site. For both panels, $r$ denotes the Pearson correlation coefficients between the measured and modeled long-term trend in the annually averaged AOD

emission rate in response to climate changes almost doubles relative to the simulation with the BASE emission module, from +14 to +24% for a surface temperature change of ~3 K (corresponding to the period 2076–2100 relative to 1976–2000).

**Estimation of the direct dust–climate feedback**. We use the CMIP5 and CESM simulations to estimate the value of $\kappa$, which we then combine with constraints on the dust DRE[1] to estimate the direct dust–climate feedback. For the CMIP5 simulations, we find a wide distribution of $\kappa$ values, with a median of 0.013 K$^{-1}$ and a 95% confidence interval (CI) spanning from −0.053 to 0.073 K$^{-1}$ (Fig. 3). Although the majority (14 out of 18) of models predict an increase in future dust loading, CMIP5 simulations include land use changes[36], which we are unable to correct for and which skews $\kappa$ to more positive values. The contribution of these land use changes to simulated future dust loading changes will depend on many factors, including the details of each model's dust emission parameterization. However, an analysis by Ward et al.[30] suggested that future land use changes will increase dust emissions by ~6% by the end of the century under the RCP8.5 scenario. Since this is similar to the median dust loading changes predicted by the CMIP5 model ensemble[29], it remains unclear from the CMIP5 simulations whether dust loading will increase or decrease in the future.

In contrast to the ensemble of CMIP5 simulations, the CESM simulations do not include land use changes, such that the simulated future changes in the global dust emission rate arise from changes in climate and $CO_2$ concentration. The CESM simulation with the BASE emission module yields $\kappa = 4.3 \pm 0.3\%$ K$^{-1}$ ($2\sigma$), which is within the 95% CI of the dust cycle response to climate changes simulated by the CMIP5 ensemble (Fig. 3). However, CESM with the PHYS dust emission module yields $\kappa = 7.7 \pm 0.4\%$ K$^{-1}$, which exceeds the 95% CI of the CMIP5 ensemble.

We combined these simulated values of $\kappa$ with constraints on the present–climate dust DRE (see Methods) in order to estimate the order of magnitude of the global direct dust–climate feedback $\lambda_{dust}$ in Eq. (4). For the CMIP5 ensemble, the wide probability distribution of $\kappa$ yields a correspondingly wide distribution for the direct dust–climate feedback: $\lambda_{dust} = -0.001$ (−0.023 ±0.013) Wm$^{-2}$ K$^{-1}$ (Fig. 4a). CESM with the BASE dust module yields $\lambda_{dust} = -0.007$ (−0.023 to +0.009) Wm$^{-2}$K$^{-1}$, which is again consistent with the range suggested by the CMIP5 simulations. In contrast, using the more physically based K14 dust module enhances the dust cycle response to future climate change (Figs. 2, 3), and thus

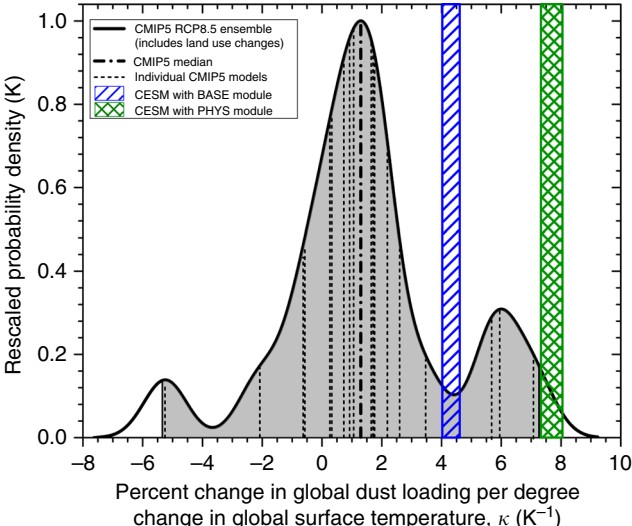

**Fig. 3** Estimates of change in global dust loading per degree globally-averaged surface temperature change. Vertical dashed lines denote individual CMIP5 model estimates of $\kappa$[29], which were used to obtain the probability density (black line; gray shading denotes the CI) using kernel density estimation[50]. The blue and green hatched boxes, respectively, denote $\kappa$ with CI for the CESM simulations with the BASE and PHYS dust emission modules. These simulations suggest that $\kappa$ might be in the approximate range of −5 to +10% K$^{-1}$

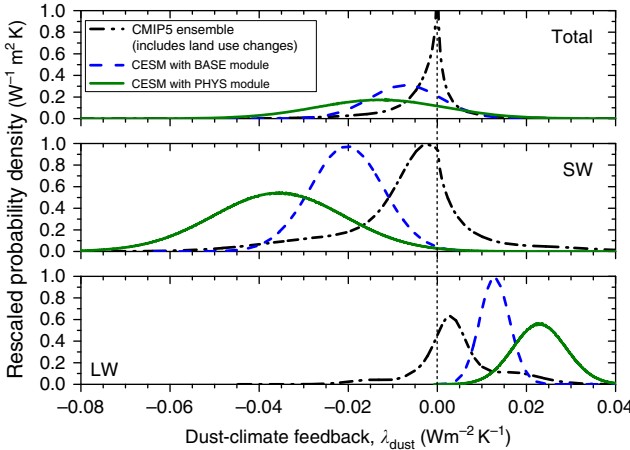

**Fig. 4** Estimates of the global direct dust–climate feedback. Results are calculated from the ensemble of CMIP5 simulations (black dash-dotted lines), and from CESM simulations with the BASE (blue dashed lines) and PHYS (green solid lines) emission modules. The total feedback (top) could equal a substantial fraction of the climate system's total direct aerosol feedbacks[29]. The direct dust–climate feedback ($\lambda_{dust}$) consists of somewhat larger opposing contributions from cooling SW interactions (middle) and warming LW interactions (bottom)

yields a larger direct dust–climate feedback of −0.013 (−0.041 to +0.016) Wm$^{-2}$K$^{-1}$ (Fig. 4a), which is partially outside of the range suggested by the CMIP5 simulations. These three estimates of the direct dust–climate feedback are themselves a sum of opposing feedbacks, due to changes in cooling interactions with SW radiation and warming interactions with LW radiation. We find that the feedback due to only SW cooling interactions is in the range of −0.06 to +0.02 Wm$^{-2}$K$^{-1}$ (Fig. 4b), whereas that due to counteracting LW warming is approximately half that at −0.01 to +0.03 Wm$^{-2}$K$^{-1}$ (Fig. 4c).

Close to source regions, the regional feedback $\tilde{\lambda}_{dust}$ is substantially greater than the globally averaged direct dust–climate feedback. We estimate $\tilde{\lambda}_{dust}$ by combining our estimates of $\kappa$ with an ensemble of climate model simulations of how the global dust DRE[1] is distributed across the globe (see Methods and Supplementary Figs. 6, 7). We find that the regional direct dust–climate feedback is enhanced over its globally averaged value (Eq.(6)) by about an order of magnitude close to major source regions, including in Northern Africa, the Sahel, the Mediterranean region, the Middle East, and Central Asia (Supplementary Fig. 8). However, the value of $\tilde{\lambda}_{dust}$, and even its sign, is highly uncertain, owing to large uncertainties in $\kappa$ (Fig. 3), the global dust DRE[1], and in how that global DRE is distributed across the globe (Supplementary Fig. 6). Figure 5 provides an estimate of that large uncertainty by showing the regional dust–climate feedback obtained from the median value of $\kappa$ from the CMIP5 ensemble (Fig. 5a), its lower CI (Fig. 5b), its upper CI (Fig. 5c), and from the CESM simulation using the PHYS module (Fig. 5d). Despite the large uncertainty in $\tilde{\lambda}_{dust}$, we find that its magnitude is likely of the order of one to several tenths of Wm$^{-2}$K$^{-1}$ close to source regions.

## Discussion

By combining constraints on the global dust DRE with a series of simulations of the global dust cycle's response to climate changes (Fig. 3), we find that the global direct dust–climate feedback lies in the approximate range of −0.04 to +0.02 Wm$^{-2}$K$^{-1}$ (Fig. 4a). This suggests that the direct dust feedback is of the order of 1% of the climate system's ~2 Wm$^{-2}$K$^{-1}$ of total physical feedbacks[25]. Moreover, dust might account for a substantial fraction of the direct climate feedback of approximately −0.02 to −0.09 Wm$^{-2}$K$^{-1}$ from all aerosols[4,29]. The sign of the global direct dust–climate feedback remains unknown, both because it is unclear whether dust emissions will increase or decrease in response to future climate changes (Fig. 4), and because the sign of the dust DRE remains uncertain[1].

Our results indicate that the sign and magnitude of the direct dust–climate feedback varies greatly on regional scales (Fig. 5). This occurs in part because of large spatial variability in dust loading, and in part because the global dust DRE is the sum of counteracting cooling and warming effects that are both modulated by a variety of factors. Dust cooling effects arise from scattering of SW radiation, which dominates for fine dust and is enhanced over dark surfaces[37]. In contrast, dust warming effects arise from scattering of LW radiation and absorption of SW and LW radiation[1]. These warming effects dominate for coarse dust and are enhanced over bright surfaces and for high altitude dust layers[1,37]. The dependence of the radiative effects on particle size causes the fining of the dust size distribution during long-range transport to produce a gradual shift from warming interactions to cooling interactions. However, this effect is often overwhelmed by the coincident shift from high albedo deserts close to source regions to low albedo ocean and vegetated surfaces further from source regions (Fig. 5 and Supplementary Fig. 6). Indeed, we find that the regional direct dust–climate feedback over very bright surfaces (the high albedo deserts of North Africa and the Middle East, and ice or snow-covered regions) is of opposite sign to that over dark surfaces (primarily oceans and vegetated regions) (Fig. 5 and Supplementary Figs. 6, 7). This causes the dust DRE and climate feedback to display a dipole pattern between the

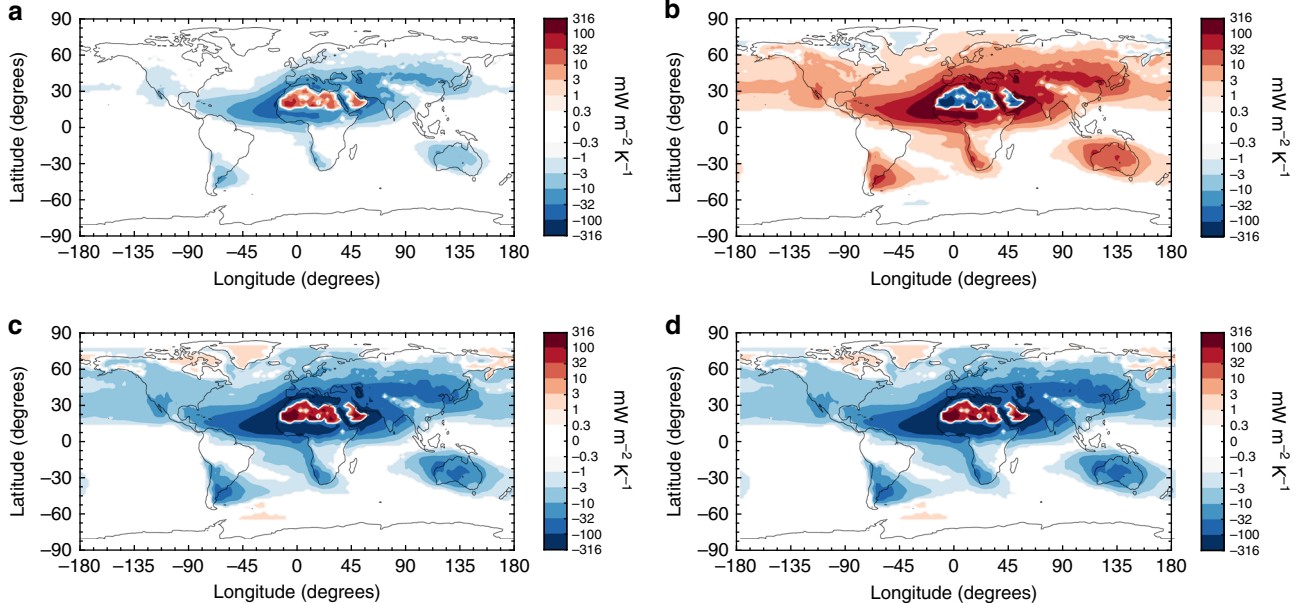

**Fig. 5** Estimates of the regional direct dust–climate feedback. **a–c** show the direct dust–climate feedback ($\tilde{\lambda}_{dust}$) obtained using the median, the lower 95% CI, and the upper 95% CI of the global dust loading response to surface temperature change ($\kappa$) from CMIP5 simulations, respectively. **d** shows $\tilde{\lambda}_{dust}$ using $\kappa$ from the CESM simulation with the PHYS emission module. Since the probability distribution of $\kappa$ derived from the CMIP5 simulations spans both negative and positive values, the sign of the dust climate feedback in **b** is opposite that of the other panels. These results thus indicate both the large uncertainty in $\tilde{\lambda}_{dust}$, and that its value could be substantial near source regions

highly reflective deserts of North Africa and the Middle East, and their surrounding oceanic and vegetated regions. This dipole does not emerge for other source regions, such as the Australian and Chinese deserts, because those source regions are less reflective. In addition to these regional variations in its sign, the magnitude of the direct dust–climate feedback is greatly enhanced close to source regions, where dust loading is much greater than the global average (Supplementary Fig. 8). In particular, we find that the direct dust–climate feedback is of the order of one to several tenths of a $Wm^{-2}K^{-1}$ in North Africa, the Sahel, the Mediterranean, the Middle East, and Central Asia (Fig. 5).

These results suggest that the direct dust–climate feedback will play an important role in shaping the future climate near major source regions, as has likely occurred in the past[38]. In idealized modeling experiments, the equilibrium surface and atmospheric temperature response to dust direct radiative forcing scales with the TOA forcing[39]. Thus, the direct forcing patterns in Fig. 5a and d imply enhanced surface and atmospheric warming over the Saharan and Arabian deserts, where there is already an amplified response to global warming[40]. Over water, the cooling is strongest in the tropical North Atlantic and over the Arabian Sea. In the North Atlantic, the meridional asymmetry in the forcing would likely excite a coupled response—the Atlantic Meridional Mode—resulting in a southward shift of the Intertropical Convergence Zone, thus weakening the West African Monsoon[41]. This forced change in the Atlantic Meridional Mode[42] and the increased dust loading[43] would both conspire to inhibit tropical cyclone formation and intensification over the North Atlantic. Over the Arabian Sea, the dust forcing would weaken the climatological northward meridional sea surface temperature (SST) gradient and amplify the cooling associated with absorbing aerosols emitted from the Indian Subcontinent, weakening the monsoonal circulation and possibly improving conditions for the formation of very strong tropical cyclones there[41]. The warming over the Sahara would increase the land–sea temperature gradient, which could strengthen the summer monsoon's northward penetration into the continent[43]. Indeed, it is thought that the enhanced rainfall of the African Humid Period (AHP), which was a period

of 15–5 kyr BP when the Sahara was almost completely vegetated and supported perennial lakes, was the result of orbital forcing that similarly enhanced the land–sea temperature gradient[44]. However, studies have also shown that the enhanced monsoonal rainfall of the AHP was also encouraged by a reduction in the concentration of dust over the continent[45], as heating by an elevated dust layer reduces rainfall by increasing atmospheric static stability[44]. Thus, the impact on the West African Monsoon may be a balance between the partitioning of dust atmospheric heating and surface forcing.

Our results distinctly include the possibility that dust loading might increase in response to future warming (Fig. 3). This would be opposite to the response inferred from paleoclimate, since colder climates tend to be dustier over the last million years (e.g., the last glacial maximum is substantially dustier than the mid-holocene warm period)[7,9,10,38]. This discrepancy suggests either that feedbacks relevant on longer geologic timescales are different than the feedbacks acting on the century timescale modeled here, or that the feedbacks are sensitive to the initial state as indicated by ice core dust records[9], and thus that the dust–climate feedbacks over the coming decades to centuries is not well constrained by paleoclimate feedbacks over the last million years.

Although it remains an open question whether global dust loading will increase or decrease in the future, our results suggest that current climate models might underestimate the dust cycle's response to climate change. In particular, we find that accounting for the additional physics included in the K14 emission scheme enhances the sensitivity of the dust cycle to climate changes, which might have contributed to an improved ability to reproduce observed long-term changes in dust AOD (Fig. 1). Specifically, the K14 parameterization accounts for the increase in the emitted vertical dust flux per unit of horizontal saltation flux that occurs when soils become more erodible, which for instance arises from the predicted future drying of soils on most deserts margins[16,35] (Supplementary Fig. 5). The net effect of accounting for this and other additional physics is a near-doubling of the increase in dust emissions per degree surface temperature change (Fig. 2), causing it to exceed the range spanned by CMIP5 models

**Table 1 Overview of the CESM/CAM4/CLM4-CN simulations used in this study**

| Simulation | CESM set-up | Dust flux parameterization | Dust source function | Meteorological data used | Time period | Objective |
|---|---|---|---|---|---|---|
| BASE-AVHRR | CLM4-CN coupled to CAM4 | Zender et al.[19] | Zender et al.[51] | MERRA | 1980–2008 | Compare against AVHRR record |
| BASE-AERONET | CLM4-CN coupled to CAM4 | Zender et al.[19] | Zender et al.[51] | ERA-I | 1994–2011 | Compare against AERONET record |
| BASE-Future | CLM4-CN | Zender et al.[19] | Zender et al.[51] | ECHAM5 CMIP3 | 1976–2100 | Estimate dust-climate feedback |
| PHYS-AVHRR | CLM4-CN coupled to CAM4 | Kok et al.[17] | None | MERRA | 1980–2008 | Compare against AVHRR record |
| PHYS-AERONET | CLM4-CN coupled to CAM4 | Kok et al.[17] | None | ERA-I | 1994–2011 | Compare against AERONET record |
| PHYS-Future | CLM4-CN | Kok et al.[17] | None | ECHAM5 CMIP3 | 1976–2100 | Estimate dust-climate feedback |

All simulations were performed at a resolution of 1.9° × 2.5°

(Fig. 3). This result suggests that empirical dust emission parameterizations in current climate models, which do not account for the increase in dust emissions per unit saltation flux that occurs as soils become more erodible, and instead generally use dust source functions to parameterize spatial variability in emissions[18,21], might cause some models to underestimate the response of the global dust cycle to climate changes.

The uncertainty in the response of the global dust loading to future climate changes contributes to large uncertainties in the direct dust–climate feedback on both global and regional scales (Figs. 4, 5). Although our methodology quantifies and propagates this uncertainty, it has a number of important limitations that nonetheless likely cause a substantial underestimation of the uncertainty in the direct dust–climate feedback (see Methods). This arises because the direct dust–climate feedback depends on a wide range of processes in the Earth system, many of which remain uncertain[13–15], and this study is able to quantify the uncertainty for only a subset of these processes (see Methods). Furthermore, the model projections of future dust loading changes (Fig. 4) might be affected by large biases in representing one or more of the processes that drive changes in the dust cycle, which are not quantified by the uncertainties propagated here. The presence of such biases is indicated by the inability of CMIP5 models to reproduce historical changes in dust[27], even when forced by observed SSTs, although this discrepancy might be partially due to inaccuracies in representing the evolution of land use changes, which are difficult to quantify[46].

Our study focuses on the feedback due to direct radiative interactions only. As such, it does not address the many indirect effects that contribute to the dust–climate feedback, and which could potentially be of substantially greater magnitude than the direct dust–climate feedback[3,47]. In particular, future studies should account for the effects of dust interactions with clouds (especially ice clouds), biogeochemistry, and the cryosphere. Interactions with biogeochemistry likely result in a drawdown of $CO_2$ and thus produce a net cooling effect[6], whereas dust-cryosphere interactions warm by decreasing the surface albedo[2]. Dust interactions with clouds induce a variety of indirect effects with opposing signs, such that it remains unclear whether dust-cloud interactions warm or cool the Earth system[3,47,48]. This lack of quantitative understanding of dust indirect effects is especially problematic for understanding dust–climate feedbacks at high latitudes, where dust–cryosphere and dust–cloud interactions are likely to dominate over direct radiative effects[49]. Overall, a substantial body of further research is needed to reduce the uncertainty on the magnitude and the sign of the dust–climate feedback on both global and regional scales.

This paper provided the first explicit estimation of the dust–climate feedback due to changes in the direct radiative effect. We did so by using a simple theoretical framework for the direct dust–climate feedback (Eq. 4) that combines recent constraints on the global dust DRE[1] with a series of model simulations of the dust cycle's response to climate changes. A subset of these simulations used a recent physically explicit dust emission parameterization[17], which increased model skill in reproducing past changes in dust emissions (Fig. 1), thereby lending some limited confidence in predictions of the dust cycle response to future climate changes (Fig. 2). A major advantage of our approach over simply using ensembles of climate model results is that it allows for both the explicit inclusion of experimental and observational constraints on dust properties and abundance[1], and the propagation of many (but not all; see Methods) of the uncertainties that affect the direct dust–climate feedback (Figs. 3, 4). Our results suggest that global dust loading could change by about −5 to +10% per degree of globally averaged surface temperature increase (Fig. 3), resulting in a global direct dust–climate

feedback in the range of $-0.04$ to $+0.02$ $Wm^{-2}K^{-1}$ (Fig. 4). As such, the global direct dust–climate feedback is likely of the order of one percent of the total physical feedbacks in the climate system[25], and could account for a large fraction of the total aerosol feedbacks[29]. Since the direct dust–climate feedback can be enhanced by over an order of magnitude close to source regions (Fig. 5), it could play an important role in shaping the future climates of North Africa, the Mediterranean, the Sahel, the Middle East, and Central Asia.

## Methods

**CMIP5 simulations of future changes in global dust loading.** We used CMIP5 simulations[28], driven by the RCP8.5 scenario[36], of the relative change in the global dust loading for the period 2090–2099 versus 2006–2015 (Table S1 of ref. [29]). We obtained $\kappa$, the relative change in global dust loading per degree surface temperature change, by normalizing these values by the temperature change over this period (Robert Allen, personal communication, 2017). This yielded 18 values of $\kappa$, from which we calculated the probability density function (black line in Fig. 3) using kernel density estimation with a Gaussian kernel with standard smoothing parameter following Eq. (3.31) in Silverman[50].

**CESM simulations of climate-induced dust changes.** Dust emissions in CESM are calculated by the CLM. We used CLM version 4.0, extended with a carbon–nitrogen biogeochemical model (CLM4-CN)[24]. To calculate dust emissions, the three BASE simulations (Table 1) used CLM's default dust flux parameterization[8,19], which uses a preferential dust source function[51] to calculate the total dust emissions.

The three PHYS simulations instead used the more physically based dust emission parameterization K14[17] (Table 1), which has a number of improvements over previous parameterizations. First, K14 accounts for the increasing scaling of dust flux with wind speed that occurs when a soil becomes less erodible (i.e., its threshold friction velocity $u_{*t}$ increases), which in CLM is primarily due to changes in soil moisture. Second, K14 explicitly accounts for the increase in dust emission efficiency that occurs as a soil becomes more erodible. That is, if a soil's $u_{*t}$ decreases, for instance due to a decrease in soil moisture, it will on average increase the amount of dust emitted per unit of horizontal saltation flux because dust aggregates in the soil will generally (but now always) be less strongly bound. Since this improvement increases the sensitivity of emissions to $u_{*t}$, it shifts those emissions to the world's most erodible regions, where $u_{*t}$ is lowest. This eliminates the need to use a source function to shift emissions to those regions, at least in CESM[21]. The fact that K14 does not require a source function is an important advantage, because using a source function to parameterize the spatial variability of dust emissions limits a model's ability to capture changes in dust emissions in a climate that differs from that for which the source function was obtained[21]. These improvements cause the K14 parameterization to produce better agreement with small-scale dust emission measurements, although substantial differences remain[17]. Furthermore, when implemented in CESM, K14 improves agreement with the global pattern of dust AOD as measured by AERONET sun photometers[21].

For both dust modules, the effects of soil moisture on the dust emission threshold friction velocity, $u_{*t}$, follows the parameterization of Fécan et al.[52], the accuracy of which was evaluated for CESM in ref. [21]. Note that this treatment differs from the default CESM parameterization, which also uses Fécan et al.[52], but uses a tuning parameter introduced in ref. [19] that practically eliminates the sensitivity of dust emissions to soil moisture in CESM. Restoring the Fécan et al.[52] parameterization to its original form also improves model agreement with the spatial distribution and seasonal and daily variability of AERONET AOD measurements[21]. Furthermore, using the original Fécan et al.[52] soil moisture sensitivity causes CESM to better reproduce historical changes in dust AOD (not shown).

In addition to wind speed and soil moisture, dust emissions are also affected by the presence of vegetation. The dust emission flux in CESM is parameterized to decrease linearly with the leaf area index (LAI), going to zero at a threshold LAI of 0.3[6]. Future and historical simulations are driven with land cover that is representative of the year 2000. As such, although seasonal and climatic changes in LAI are prognostically modeled in CLM4-CN, this model does not account for possible changes in the biogeography due to changes in land use or climate. We chose this approach because historical changes in the West African dust plume (Fig. 1a) are driven almost entirely by changes in wind speed[11], and because estimating the dust climate feedback requires the isolation of the dust cycle's response to changes in climate (see discussion in main text).

For both the BASE and the PHYS dust emission models, we ran two separate CESM simulations to evaluate the model's ability to reproduce historical changes in dust emissions. The first pair of simulations, BASE-AVHRR and PHYS-AVHRR, model the global dust cycle over the historical period 1980–2008, with the objective of evaluating the model's ability to reproduce the historical trend in aerosol optical depth (AOD) over the tropical North Atlantic ocean, derived from the long-term AOD record obtained by the Advanced Very High Resolution Radiometer (AVHRR)[32]. The second pair of simulations, BASE-AERONET and PHYS-

AERONET, model the global dust cycle over the historical period 1995–2011 with the objective of evaluating the model's ability to reproduce long-term trends in AOD measurements by AERONET stations. The BASE-AERONET and PHYS-AERONET simulations have been described in detail in ref. [21], and the BASE-AVHRR and PHYS-AVHRR simulations use the same model set-up, with the modifications described above. All four simulations used CESM version 1.1[53], and the first year of each simulation was used as model spin-up and not used for analysis.

In both pairs of simulations, the dust emissions calculated by CLM4-CN were used by CESM's atmosphere model, the Community Atmosphere Model version 4 (CAM4), to calculate the three-dimensional transport and deposition of dust, as well as the dust aerosol optical depth[8,54]. To do so, CAM4 distributes the emitted dust into four size bins between 0.1 and 10 μm diameter following a size distribution parameterization based on brittle fragmentation theory[55]. This parameterization is in good agreement with a wide range of measurements[2] and does not depend on the wind speed at emission, which is consistent with measurements[56]. The optical properties for each bin are specified as in ref. [54]. CAM4 accounts for dry and wet deposition of dust and includes the effects of other aerosol species.

The simulations used the capability of CESM to be forced with reanalysis winds instead of predicted winds. Specifically, the BASE-AVHRR and PHYS-AVHRR runs used the MERRA reanalysis meteorology from the Goddard Earth Observing System of the NASA Global Modeling and Assimilation Office (see https://gmao.gsfc.nasa.gov/reanalysis/MERRA/), which extends back to the start of the AVHRR record in 1982. However, the use of the ERA-Interim reanalysis meteorology[57], which only extended back to 1989 and could thus not be used for the AVHRR runs, results in more realistic dust emissions[58]. Therefore, the BASE-AERONET and PHYS-AERONET simulations used the ERA-Interim reanalysis. However, instead using the MERRA reanalysis similarly yields a better agreement of the PHYS-AERONET simulation against the AERONET data than does the BASE-AERONET simulation.

For each simulation, the global tuning parameter that scales the dust emissions was set in such a way that it eliminates the bias with AERONET AOD measurements in dusty regions[21].

**Observations to evaluate modeled dust emission changes.** We used the results of the two pairs of historical simulations to evaluate the ability of the new K14 dust emission scheme to reproduce past changes in dust emissions. We thus focused this evaluation on measurements close to dusty regions, in order to have confidence that observed and modeled changes are largely due to changes in dust emissions, and not due to changes in circulation patterns or depositional processes. As such, we compared the simulations against satellite-derived estimates of dust optical depth off the West African coast around Cape Verde[27]. Since this record extends back to 1982, it allows us to test whether the model reproduces variability in dust emissions from the world's main source region over the past several decades, when forced by winds from reanalysis meteorology.

We also compared the simulations against long-term changes in AOD at eight AERONET stations[33] for which such long-term changes were dominated by changes in dust. That is, we selected AERONET stations that both have long data records, and for which we can be confident that long-term changes in AOD are due to changes in dust, as opposed to changes in other aerosols. We therefore selected stations with at least 5 years for which an annual mean AOD can be defined, subject to the further quality control criteria below. Since smaller values of the Angstrom exponent $\alpha$ indicate coarser aerosols[1,2], we use measurements of $\alpha$ in the 440–870 nm wavelength range to select stations for which long-term AOD changes are due to dust. This can be indicated either by (i) very low values of $\alpha$, indicating that AOD is overwhelmingly due to dust, such that changes in AOD are very likely due to changes in dust, or (ii) by increases in AOD being associated with a coarsening of aerosols and thus a decrease in $\alpha$. Correspondingly, we selected stations that either have a very low Angstrom exponent, $\alpha < 0.4$, or for which the majority of the variance in dust AOD is explained by changes in the Angstrom exponent, as quantified by a correlation between $\alpha$ and AOD of $r < -0.70$. To exclude the possibility that changes in other coarse aerosols, notably sea salt, are driving changes in AOD, we further only use stations for which over 75% of the AOD is due to dust, based on the simulations reported in Kok et al.[21].

The above procedure resulted in the selection of eight AERONET stations with long-term trends driven by changes in dust: five in North Africa, one in the North Atlantic, and two in the Middle East. We processed the data at each of these stations to obtain the annually averaged AOD, and to remove the effects of changes in the exact days on which AOD was measured between years at a given site. First, we calculated the monthly averaged AOD for months with at least 5 days of data. For this, we used the daily averaged level 2.0 quality assured AOD (pre-field and post-field calibrated and manually inspected), obtained from the Version 2 Direct Sun Algorithm. We then used these monthly averages to calculate the seasonal AOD cycle at each station from the average of all available monthly averaged AOD values for each of the 12 months (see Supplementary Fig. 2). We then subtracted this seasonal AOD cycle from each monthly averaged AOD to obtain the anomaly in the AOD for each month. To obtain the annually averaged AOD for each year, we then averaged this anomaly over all 12 months in a given year, and then added this annually averaged AOD anomaly to the mean AOD obtained from averaging over the seasonal cycle of AOD.

We followed a similar procedure to calculate the annually averaged modeled dust AOD for each site. That is, we used the simulated AOD at the exact days for which measurements were available to first calculate the modeled monthly averaged AOD, which we then averaged for each of the 12 months to obtain the seasonal AOD cycle. We then subtracted this from each modeled monthly averaged AOD to obtain the AOD anomaly, which was then averaged over all months in a given year to obtain the modeled annually averaged AOD anomaly. We then obtained the modeled annually averaged AOD by adding this anomaly to the mean AOD obtained from averaging over the modeled seasonal AOD cycle. The resulting comparison between measured and modeled AOD is plotted in Supplementary Figure 1.

**CESM simulations of dust response to future climate changes**. The BASE-future and PHYS-future simulations use CLM4-CN to simulate the response of the global dust emission rate $Q$ to climate changes until the year 2100. We forced CLM4-CN with a preindustrial atmospheric $CO_2$ concentration (285 ppmv) until it reached equilibrium. Specifically, we cycled the first 25 years (1948–1972) of the Qian et al.[59] NCEP/NCAR reanalysis data set (temperature, precipitation, winds, etc.), and repeated this procedure until the 25-year average of the net ecosystem exchange fluxes fell below 0.05 PgC/year[60]. After CLM4-CN was spun up in this manner, we conducted model experiments from the year 1798 until 1972 following the procedure outlined in ref. [61]. That is, we cycled the 1948–1972 Qian et al.[59] reanalysis data, combined with historical reconstructions of $CO_2$ concentration[62], nitrogen deposition[63], and land cover change[64]. From 1973 onward, we forced the model with the same 1948–1972 repeat cycle, but with climate anomalies added, and with the $CO_2$ concentration increasing following the SRES (Special Report on Emissions Scenarios) A1B scenario. These climate anomalies were taken as the average difference between the monthly mean of fully coupled simulations with the ECHAM5/MPI-OM model[65] and the 1948–1972 reanalysis, and are described in more detail in ref. [61]. The 25-year cycling of reanalysis data was done to create realistic interannual variability in the simulated future climate, which is important because of the non-linear dependence of dust emissions on wind speed, soil moisture, and vegetation. We consequently report model results in terms of 25-year running averages. We used output from ECHAM5, because Mahowald[14] showed that this model predicts changes in the global desert area that are characteristic of the mean of the models that participated in Coupled Model Intercomparison Project phase 3 (CMIP3). The ECHAM5 simulations were forced with the A1B scenario, as part of the CMIP3. Since the historical reconstructions of nitrogen deposition and land cover change only extend until 2005[63,64], and since we desired to separate the response of the dust cycle to climate changes from other anthropogenic changes such as land use, we kept nitrogen deposition and land use from 2006 onward constant at 2005 levels. Note that human land use changes can also substantially impact dust emissions[46]. Since such changes are the direct result of human actions, they constitute an anthropogenic forcing, not a feedback, and are thus not included in our simulations. For both the BASE-future and PHYS-future simulations, the global tuning constant for dust emissions was set such that the global dust emission rate over the period of 1995–2011 equaled that for the same period for the BASE-AERONET and PHYS-AERONET simulations, respectively, for which the global tuning constant was calibrated to AERONET observations (see above and ref. [21]).

We used the BASE-future and PHYS-future simulations of the change in the global dust emission rate to obtain $\kappa$. Specifically, we use the global emission rate as a proxy for the global dust loading, such that we approximated $\kappa$ as the relative change in dust loading ($\Delta Q / Q_0$) per degree surface temperature change ($\Delta T$), which we obtained from the linear least-squares fits of $\Delta Q / Q_0$ to $\Delta T$ (Fig. 3). We use this assumption because previous model simulations have indicated that the modeled dust lifetime, which determines how the emission rate is converted to loading, is relatively insensitive to climate[6]. Furthermore, uncertainties in other components of the estimation of $\kappa$ and $\lambda_{dust}$, such as due to uncertainties in future precipitation change and the dust DRE, are quite large, such that uncertainties due to possible changes in dust lifetime is likely to be a small contributor to the uncertainty in $\kappa$.

**Estimation of the direct dust–climate feedback**. We estimated the global direct dust–climate feedback by combining the CMIP5 and CESM estimates of $\kappa$ (Fig. 3) with the constraint on the present-day dust DRE ($\zeta_0$), and its SW ($\zeta_{0,SW}$) and LW ($\zeta_{0,LW}$) components, from Kok et al.[1]. This recent study constrained $\zeta_0$ to $-0.20$ ($-0.48$ to $+0.20$) Wm$^{-2}$ by combining an analysis of the size-resolved dust loading with four climate model simulations of the radiative effect per unit dust aerosol optical depth. We simplified these results as a normal distribution with mean and standard deviation equal to those of the full DRE probability distributions given in Supplementary Figure 5 of Kok et al.[1], yielding $\zeta_0 = -0.17 \pm 0.19$ Wm$^{-2}$, $\zeta_{0,SW} = -0.47 \pm 0.19$ Wm$^{-2}$, and $\zeta_{0,LW} = +0.30 \pm 0.08$ Wm$^{-2}$. Combining these probability distributions with that for $\kappa$ (Fig. 3) then yielded the probability distributions of $\lambda_{dust}$ (Eq. 4 and Fig. 4).

We obtained the regional direct dust–climate feedback, $\tilde{\lambda}_{dust}$, by combining estimates of $\kappa$ (Fig. 3) with constraints on the regional dust DRE, $\tilde{\zeta}_0$. In turn, we obtained $\tilde{\zeta}_0$ by combining the global dust DRE[1] with an ensemble of four climate model simulations[66] of how that global dust DRE is distributed regionally. Specifically, for each model $i$, we obtained the simulated present climate dust DRE

due to both SW and LW interactions for each modeled particle bin $k$ as a function of location. Since the dust DRE generated by a particle bin scales with that bin's dust AOD[37], we multiplied each model bin's SW and LW DRE with a normalization factor $\alpha_{i,k}$ that corrects the simulated globally averaged dust AOD for that particle bin to the constraint on the globally averaged dust AOD obtained in Kok et al.[1]. That is, the DRE due to SW $\left(\tilde{\zeta}^i_{0,SW}(\theta,\phi)\right)$ and LW $\left(\tilde{\zeta}^i_{0,LW}(\theta,\phi)\right)$ interactions for climate model $i$ is calculated as:

$$\tilde{\zeta}^i_{0,SW}(\theta,\phi) = \frac{1}{N_P} \sum_{p=1}^{N_P} \sum_{k=1}^{N_{k,i}} \alpha_{i,k} \tilde{\zeta}^i_{0,SW,k}(\theta,\phi), \quad (7)$$

$$\tilde{\zeta}^i_{0,LW}(\theta,\phi) = \frac{1}{N_P} \sum_{p=1}^{N_P} \sum_{k=1}^{N_{k,i}} \alpha_{i,k} \tilde{\zeta}^i_{0,LW,k}(\theta,\phi), \quad (8)$$

where $\tilde{\zeta}^i_{0,SW,k}(\theta,\phi)$ and $\tilde{\zeta}^i_{0,LW,k}(\theta,\phi)$ are the DREs predicted by model $i$ for particle bin $k$ at longitude $\theta$ and latitude $\phi$, due to interactions with SW and LW radiation, respectively, and $N_{k,i}$ denotes the number of particle bins simulated by climate model $i$. This calculation is averaged over a large number ($N_P = 10^4$) of realizations of the correction factor $\alpha_{i,k}$, which equals

$$\alpha_{i,k} = \frac{\hat{\tau}_{d,k}}{\tau_{d,i,k}}. \quad (9)$$

Here $\tau_{d,i,k}$ is the DAOD predicted by model $i$ for particle bin $k$, and $\hat{\tau}_{d,k}$ is a realization of the global DAOD due to particle bin $k$ drawn from the probability distribution obtained in ref. [1] (see their Figs. 2c and S1). The SW and LW DRE that result from the above procedure are plotted in Supplementary Fig. 6 for each of the four climate models used.

We then obtained the regional dust SW $\left(\tilde{\zeta}_{0,SW}\right)$ and LW $\left(\tilde{\zeta}_{0,LW}\right)$ DRE by averaging over the SW and LW DRE obtained for each of the $N_i = 4$ climate model simulations in the model ensemble[66],

$$\tilde{\zeta}_{0,SW}(\theta,\phi) = \frac{1}{N_i} \sum_{i=1}^{N_i} \tilde{\zeta}^i_{0,SW}(\theta,\phi), \quad (10)$$

$$\tilde{\zeta}_{0,LW}(\theta,\phi) = \frac{1}{N_i} \sum_{i=1}^{N_i} \tilde{\zeta}^i_{0,LW}(\theta,\phi). \quad (11)$$

The resulting SW DRE, LW DRE, and total DRE $\left(\tilde{\zeta}_0 = \tilde{\zeta}_{0,SW} + \tilde{\zeta}_{0,LW}\right)$ are plotted in Supplementary Fig. 7, and are used in the calculation of the regional direct dust–climate feedback (Eq.(5) and Fig. 5).

**Limitations of the methodology**. We list here the most important limitations of our methodology that have not already been discussed in the main text. First, the coarse resolution and incomplete description of the land surface handicaps climate models in simulating dust emissions. This occurs in large part because dust emission is a small-scale process that is sensitive to unresolved variations in both (turbulent) winds and in soil properties such as size distribution, mineralogy, soil moisture, and soil aggregation state[67]. Furthermore, dust emission can be triggered by mesoscale meteorological events that also are not captured well in many models[68], such as haboobs and the breakdown of the nocturnal low level jet. These limitations restrict the ability of models to reproduce the dust cycle even in the current climate[6,20,21], and thus adds substantial uncertainty to simulations of the dust cycle in a future climate. Therefore, although we use an ensemble of climate models to constrain $\kappa$, it is possible that climate models have systematic biases in simulating dust cycle response to climate changes, which cannot be quantified here. Second, the CMIP5 and CESM simulations do not account for changes in the dust emission rate due to changes in source regions, other than those already captured in its physically based scheme. That is, these simulations do not account for (i) changes in (fluvial) sediment supply and thus erodibility of existing source regions, and (ii) changes in biogeography, i.e., changes in biomes rather than changes in LAI within a biome, which CESM does account for. Third, the simulated response of the global dust cycle to climate changes is known to be highly sensitivity to both the treatment of the response of vegetation to changes in climate and $CO_2$ concentration, and to future changes in precipitation and surface temperature[14,15]. Whereas the ensemble of CMIP5 simulations captures a range of possibilities for both future changes in meteorology and vegetation response to changes in climate and $CO_2$, the CESM future simulations do not. That is, these simulations were forced with atmospheric simulations that are close to the median of the CMIP3 climate model ensemble, and use only a single parameterization of vegetation response to climate and $CO_2$[24]. Fourth, the CMIP5 and CESM simulations, respectively, use the RCP8.5 and A1B scenarios, and we did not test the sensitivity of the direct dust–climate feedback to the emissions scenario. Nonetheless, because the direct dust–climate feedback is implicitly normalized by the surface temperature change, we expect differences in the direct dust–climate feedback between scenarios to be substantially less than they would be for model calculations of dust

radiative forcing. Fifth, the constraint on the dust DRE is subject to a number of limitations. In addition to those already described in ref.[1], recent measurements indicate that LW interactions might be less important than previously thought[69], which could cause the dust DRE to be more cooling than represented in ref. [1]. Conversely, the lifetime of coarse dust might be underestimated in ref. [1], which could have caused an underestimate of dust warming. In addition, the ensemble of simulations of the dust direct radiative effect used in Kok et al.[1] does not account for regional heterogeneity in the mineralogy of dust, which likely has a limited effect on the global DRE but does substantially affect the spatial pattern of the dust DRE[70]. A final limitation of our methodology is that, although CESM with the improved dust emission scheme and driven by ERA-Interim meteorology shows some skill in reproducing past changes in dust emissions (Fig. 1), the drivers of future changes in dust emissions are likely to differ from past ones. In particular, past changes in North African dust emissions seem to be due primarily to changes in wind speed over the Sahara[11]. However, future climate changes are expected to be due to several factors, including changes in wind speed[11], changes in desert extent due to both increasing $CO_2$ concentrations[14] and likely decreases in soil moisture in semi-arid regions[16,35], and changes in erodibility of arid regions, such as due to the decreases in soil moisture. Furthermore, since the meteorology in the PHYS-AVHRR simulations is prescribed, it does not provide information on whether CESM can correctly simulate relevant changes in wind, soil moisture, and surface temperature in a future climate, which affect the resulting dust emission. Indeed, CMIP5 models have been shown to be incapable of reproducing important changes in North African dust in the current climate when forced by observed SSTs[27], which casts substantial doubt on their ability to accurately forecast future changes in the dust cycle. The above limitations of our study suggest that our methodology underestimates the uncertainty of the direct dust–climate feedback, and thus should be seen as an order of magnitude estimate.

**Code availability**. The codes used to conduct the analysis presented in this paper and in the production of the figures are available through Github (https://github.com/jfkok/Koketal_DustClimateFeedback_NCOMMS2018_MatlabCode).

**Data availability**. CMIP5 data are available through the Earth System Grid (http://pcmdi9.llnl.gov/).CESM model simulations used here are available through Zenodo (https://zenodo.org/record/1124933), and the ensemble of global model simulations of dust DRE used here are available from Ref.[66].

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

## Acknowledgements

We acknowledge the World Climate Research Programme's Working Group on Coupled Modelling, which is responsible for CMIP, and we thank the climate modelling groups for producing and making available their model output. For CMIP the US Department of Energy's Program for Climate Model Diagnosis and Intercomparison provides coordinating support and led development of software infrastructure in partnership with the Global Organization for Earth System Science Portals. We thank Robert Allen for helpful comments and for providing the surface temperature change data of CMIP5 models. We acknowledge support from National Science Foundation (NSF) grants 1137716 and 1552519 to J.F.K. The CESM simulations were conducted at the National Center for Atmospheric Research's Computation Information Systems Laboratory, an NSF-funded facility.

## Author contributions

J.F.K. conceived the project, designed the study, conducted the CESM simulations, performed the analyses, and wrote the paper. D.S.W. and N.M.M. assisted with the design of the study and the CESM simulations, and A.T.E. provided expertise on regional climate effects. All authors discussed the results and commented on the manuscript.
