## [Peer Review File · Nature Communications]

Reviewers' comments:

Reviewer #1 (Remarks to the Author):

This is a very interesting paper, and quite unique in its (somewhat heroic) attempt to quantify the direct dust-climate feedback. For that reason alone, I think it is worthy of publication, but I have multiple questions and comments

that I would like to see addressed before that stage. Importantly, nowhere in the paper should estimates be presented as the total dust-climate feedback, given that indirect effects of dust are not considered, especially because these effects could easily dwarf the direct effect (see comment below). The paper generally does a good job specifying that it only addresses the direct dust-climate feedback, but with a few exceptions (the very end of the discussion section for example). Otherwise, I found the approach to be interesting and generally sound, although I have a few questions/concerns, as listed below. The paper is well written and generally a pleasant read. The magnitude of the global (direct) dust-climate feedback is somewhat underwhelming, so I would de-emphasize that and focus more on the regional impact. Also, as per my comment above, the feedback could strengthen significantly if effects via clouds were accounted for.

A few more specific comments:

- Line 36-37: this statement reads as if there is a direct link between CO₂ in the atmosphere and desert extent, but the impact of course occurs via changes to winds, precipitation, etc. Please rewrite.
- Line 65: "greenhouse warming" isn't appropriate here - "the positive forcing due to increasing greenhouse gas concentrations" would be better.
- This isn't correct. You're discussing the dust-climate feedback here, so the forcing is NOT due to changes in the dust direct effect. Changes to the dust direct effect would influence how much warming would be required before the TOA radiation budget is balanced after an imposed radiative forcing (i.e. it would affect dR/dT).
- Line 99-100: This needs better justification. If the word limit is an issue, you can easily shorten the somewhat lengthy justification of your definition of regional feedback above.
- Fig. 1: The improvement with the PHYS simulation is quite impressive, but what is driving this decrease in dust emissions (winds or precip?), and does it continue in the simulations of future climate, or does it reverse (consistent with the global trend). This needs further discussion/explanation.
- Line 183-184: Are these simulated increases in dustiness with warming not inconsistent with the paleoclimate record, which suggests that warm climates are LESS dusty than colder climates? Please discuss.
- Line 213-214: Were the CMIP5 kappa values corrected for land use? If not, they should not be used to calculate the dust-climate feedback.
- Line 349: There's new work out that suggests that dust-cloud interactions strongly cool climate (see <http://onlinelibrary.wiley.com/doi/10.1002/2017GL072584/abstract>) .

Reviewer #2 (Remarks to the Author):

This paper addresses an important question – what's the response of dust emissions and its direct radiative effect to a changing climate? Due to its large abundance, and strong light scattering and absorbing abilities, mineral dust may exert a strong heating or cooling radiative effects over large areas. While most early studies focused on the effects of mineral dust on the climate, very few studies have explicitly investigated the impact of climate change on dust distributions. The paper implemented a new physical-based dust emission scheme in the model and simulated the change of dust emissions with unchanged land use. The results show that the new scheme had approximately two times stronger effects of dust emission than the present scheme and thus an enhanced feedback on the direct radiative forcing. By explicitly discussing the feedback effect, this paper may help improve the understanding of aerosol-climate interactions.

The paper is well-written and provides new insights. However, I did have a couple of concerns as detailed below:

Major concern:

I appreciated the effort to explicitly decouple the feedback from complex interactions. But the dust-climate feedback has been implicitly considered in previous climate models. Extracting such existing –process is instructive but the authors need to convince the readers more for its value in climate research. For example, how is this going to change our understanding of the radiative effects of dust? From my point of view, the new dust scheme is a new aspect and hence it is crucial to demonstrate its advantage and better performance over old schemes. A physical-based scheme is not necessarily better than a well-tuned empirical approach and the authors need to provide more proofs to support this novel point. Since the radiative effect depends on the change of dust emissions as well as the absolute dust abundance (that determine the direct radiative forcing), showing the correlation coefficient of the trends is not enough and a comparison absolute dust AOD will also be required also for the available data.

Other comments:

Line 116 "most CMIP5 models are unable to reproduce historical trends in dust loading ... ", when discussing the feedback, the simulation of temperature is as important as dust, how about the performance of your CESM in reproducing the historical trends of T compared to observations and CMIP5?

Line 121 "evaluate whether the model can reproduce past changes in dust emissions", how did you treat the change of land use in simulations of past trends? Did you fix the land use as in the case of simulations for future?

Fig. 1, I would like to see a comparison of the absolute AOD as well. I also have a general question about the observations and simulations used here. How about the agreement between AVHRR and AERONET data for overlapped sites and periods of your simulations? Difference reanalysis data were used in simulations AVHRR and AERONET, how will they make a difference on the simulated absolute AOD for the overlapped period when both reanalysis data are available? Will the conclusion still hold?

Line 208, " $\kappa = 4.3$, ... which is consistent with the dust cycle response to climate changes simulated by the CMIP5 ensemble", I don't think a κ of 0.43 is consistent with a κ of 0.013 for CMIP5 ensemble. Please make it more accurate, e.g, within the 95% CI. Another question is that the base module of your CESM lies near the edge of CMIPs, could you explain why you choose this model instead of a model giving results in the middle of all CMIPs. In that case, CESM with PHYS may also produce results within 95% CI.

Fig. 5, the panels b and c look like mirror images, why?

Technical corrections:

Page 1, author list, "Ward7" should be "Ward2"

Line 62, " Feedbacks in the climate system are defined as" should be " Feedbacks in the climate system are defined according to the equation"

Line 63, please use "m⁻²" instead of "/m²"

Line 196, the unit of κ is missing

Reviewer #3 (Remarks to the Author):

The paper entitled « Estimation of the direct dust-climate feedback » presents an original work about the direct radiative effect of dust particles on climate. The authors have used recent constraints on dust loading and global climate model simulations to constrain the direct dust-climate feedback. This study brings new elements on the impact of dust aerosols in future climate, and suggests that dust aerosols could account for a significant part of the total aerosol feedbacks while they are often neglected up to now. The use of new constraints on dust loadings also enables to reduce the uncertainties and reinforces the strengthness of the results. That is why I recommend the publication of this paper in Nature Communications, after the revision of the few following minor points.

- As underlined by the authors, one of the main limitation of the paper is the absence of the indirect effect in the estimation of the dust-climate feedback. The authors could add some estimations of the quantification of indirect effect as realized in previous studies, even if the uncertainties are important. It could help the reader to understand the importance of the work on the direct effect done in this study, and what remains to do as far as the indirect effect is concerned.

- I have some difficulties to understand what the authors call the « regional dust-climate feedback », and what is really behind this estimation, as they do not consider spatial variations in the local change in dust loading per degree (lines 99-100). The authors should clarify this point.

- CMIP5 simulations: the authors should give details on the selection of models. Do they consider all the CMIP5 simulations which are available, or have they established a subset of models ? (line 108)

- Table1: is it a problem to use meteorology from a CMIP3 simulation in the future CEMS simulations compared to CMIP5 simulations for the other models ? The authors should at least give the scenario which has been used for this CMIP3 simulation.

- Have the authors considered the use of different scenarios for this study ? Would the results differ with scenarios with a less important global warming ?

- Land use change (line 110): I am not sure that all CMIP5 simulations include land use change, so maybe there is a possibility to separate two kinds of simulations ? Moreover have the others carried out also CESM simulations with land use, in order to check the impact of land use on the evolution of dust loadings ?

- Figure 1: I am surprised on the large negative bias in dust AOD in the simulation using BASE module compared to the one using PHYS module. Is there a specific tuning only in PHYS module ?

- Lines173-176: How is it possible to have both a « slight increase in wind speed over most North African dust source regions » and « a decrease in dust emissions due to North African wind speed changes » ? Is it instead an increase in dust emissions, as we can see in Fig. S3 ?

- Figure 3: I wonder if the authors have tried to understand the distribution of the different CMIP5 simulations. Is there any link between the models with a strong warming, and the models with a strong increase in dust loading ?

- Figure 5: Following my remark on regional dust-climate feedback, I am also surprised by the important difference between Fig 5a and Fig 5b, as the sign of the feedback has changed. The authors should explain more in details how you can have such changes in the sign of the feedback.

We thank both reviewers for their constructive comments, which has helped us to improve the paper. Below we include a point-by-point response to the referee comments, and describe the corresponding changes we have made to the manuscript. We hope these and other changes have addressed all comments satisfactorily.

All responses below are in blue text.

Comments from reviewer #1:

This is a very interesting paper, and quite unique in its (somewhat heroic) attempt to quantify the direct dust-climate feedback. For that reason alone, I think it is worthy of publication, but I have multiple questions and comments that I would like to see addressed before that stage.

Thank you for these positive remarks, and for your many helpful comments below.

Importantly, nowhere in the paper should estimates be presented as the total dust-climate feedback, given that indirect effects of dust are not considered, especially because these effects could easily dwarf the direct effect (see comment below). The paper generally does a good job specifying that it only addresses the direct dust-climate feedback, but with a few exceptions (the very end of the discussion section for example).

We appreciate you pointing this out, as we indeed should be very careful to avoid this confusion. We've corrected this throughout the manuscript, including at the end of the discussion section.

Otherwise, I found the approach to be interesting and generally sound, although I have a few questions/concerns, as listed below. The paper is well written and generally a pleasant read. The magnitude of the global (direct) dust-climate feedback is somewhat underwhelming, so I would de-emphasize that and focus more on the regional impact. Also, as per my comment above, the feedback could strengthen significantly if effects via clouds were accounted for.

That's a good point. We now note that explicitly in the first sentence of the second-to-last paragraph of the discussion.

A few more specific comments:

- Line 36-37: this statement reads as if there is a direct link between CO₂ in the atmosphere and desert extent, but the impact of course occurs via changes to winds, precipitation, etc. Please rewrite.

This is indeed written confusingly. We actually did mean to communicate here that CO₂ directly affects desert extent, which it does through "CO₂ fertilization", in which plant stomata can be open for less time (and thus lose less water) to obtain the same amount of CO₂. We've rewritten this sentence to more clearly separate the CO₂ fertilization effect from atmospheric warming effects on wind and precipitation (indirectly caused by rising CO₂).

- Line 65: "greenhouse warming" isn't appropriate here - "the positive forcing due to increasing greenhouse gas concentrations" would be better.

Thank you for pointing this out. We've corrected the phrasing here per your suggestion.

- This isn't correct. You're discussing the dust-climate feedback here, so the forcing is NOT due to changes in the dust direct effect. Changes to the dust direct effect would influence how much warming would be required before the TOA radiation budget is balanced after an imposed radiative forcing (i.e. it would affect dR/dT).

You are right, of course. Thank you very much for catching this erroneous wording. We've corrected the phrasing here along the lines you noted.

- Line 99-100: This needs better justification. If the word limit is an issue, you can easily shorten the somewhat lengthy justification of your definition of regional feedback above.

Model predictions of local/regional changes in dust concentration are unlikely to tell us anything useful about what the actual change will be, based on the failure of those models to predict regional historical changes per Evan et al. (2014), the large spread in future CMIP5 results, and the fact that those models also include land use changes. Consequently, explicitly using the regional predictions of CMIP5 simulations would add complexity to our methodology without, we think, increasing the accuracy of the results. We therefore consider it more logical to make the simple assumption that regional changes equal the global changes (i.e., that

$\kappa(\theta, \phi) = \kappa$). However, if we did use the regional predictions, it would be unlikely to qualitatively change our results in Fig. 5, considering the large uncertainties arising from uncertainty in kappa, the global DRE, and the regional distribution of that DRE.

To address this comment, we've expanded the discussion of this assumption in the text to better justify our choice.

- Fig. 1: The improvement with the PHYS simulation is quite impressive, but what is driving this decrease in dust emissions (winds or precip?), and does it continue in the simulations of future climate, or does it reverse (consistent with the global trend). This needs further discussion/explanation.

We've rewritten this paragraph to clarify these points, and now note that the observed decrease in dust emissions since the 1980s is likely due to the coincident decrease in wind speed over North African source regions (Supplementary Fig. 1f and Evan et al., 2016). CMIP5 models differ on whether that decrease in wind speed will continue into the next century, but the meteorology used to drive our future simulation (CMIP3 ECHAM5 meteorology) predicts a slight increase. We now discuss that a bit more clearly in the first paragraph following Fig. 1.

- Line 183-184: Are these simulated increases in dustiness with warming not inconsistent with the paleoclimate record, which suggests that warm climates are LESS dusty than colder climates? Please discuss.

This is an excellent point: the model simulations reported here and elsewhere that suggest a possible increase in future dust loading are indeed the opposite response from what is inferred from paleoclimate records. We think the most likely explanation for this is either that the dust response is sensitive to the initial state of the climate, or that the longer-term response is different from the century-scale response modeled here. As requested, we now explicitly address this point in the discussion.

- Line 213-214: Were the CMIP5 kappa values corrected for land use? If not, they should not be used to calculate the dust-climate feedback.

The CMIP5 kappa values were not corrected for land use, as discussed in the "Estimation of the direct dust-climate feedback" subsection. This is a clear weakness of using the CMIP5 simulations, which we discuss at some length in that subsection, and which we therefore list explicitly in the legend of Figures 3 and 4. There unfortunately is no model ensemble available of future dust changes that does not include land use changes, and this drawback of the CMIP5 simulations is one of the reasons (together with the availability of the improved emission scheme) that we report separate CESM simulations that do not suffer from this problem. Nonetheless, we are able to give an indication of the bias in the CMIP5 results of kappa due to the inclusion of land use changes based on the work of Ward et al. (2014), which estimated dust loading changes due to land use changes in CESM future simulations.

We've added a subclause to the "Estimation of the direct dust-climate feedback" subsection that explicitly acknowledges that we cannot correct for land use changes in the CMIP5 simulations.

- Line 349: There's new work out that suggests that dust-cloud interactions strongly cool climate (see <http://onlinelibrary.wiley.com/doi/10.1002/2017GL072584/abstract>).

Thank you for suggesting this, as this is indeed a very relevant paper. We now cite and discuss this work in the Discussion section.

Comments from reviewer #2:

This paper addresses an important question – what's the response of dust emissions and its direct radiative effect to a changing climate? Due to its large abundance, and strong light scattering and absorbing abilities, mineral dust may exert a strong heating or cooling radiative effects over large areas. While most early studies focused on the effects of mineral dust on the climate, very few studies have explicitly investigated the impact of climate change on dust distributions. The paper implemented a new physical-based dust emission scheme in the model and simulated the change of dust emissions with unchanged land use. The results show that the new scheme had approximately two times stronger effects of dust emission than the present scheme and thus an enhanced feedback on the direct radiative forcing. By explicitly discussing the feedback effect, this paper may help improve the understanding of aerosol-climate interactions.

We much appreciate your positive comments.

The paper is well-written and provides new insights. However, I did have a couple of concerns as detailed below:

Major concern:

I appreciated the effort to explicitly decouple the feedback from complex interactions. But the dust-climate feedback has been implicitly considered in previous climate models. Extracting such existing –process is instructive but the authors need to convince the readers more for its value in climate research. For example, how is this going to change our understanding of the radiative effects of dust?

This is a good point: how does the new approach, which is based partially on a simple analytical theory (Eqs. 1-6) and partially on climate model simulations, actually better inform our understanding of dust radiative effects on climate? The main advantages of our approach over simple climate model ensemble results are two-fold. First, our approach directly incorporates observational and experimental constraints on dust properties and abundance, as discussed in more detail in Kok et al. (2017). Second, our semi-analytical approach allows for the propagation of all those uncertainties that we can quantify (and we tried to be careful to point out that there are many uncertainties and model biases that we cannot quantify – see Methods), as shown explicitly through the probability distributions in Figs. 3 and 4.

To address the reviewer comment, we now communicate these points directly in a sentence added to the concluding paragraph.

From my point of view, the new dust scheme is a new aspect and hence it is crucial to demonstrate its advantage and better performance over old schemes. A physical-based scheme is not necessarily better than a well-tuned empirical approach and the authors need to provide more proofs to support this novel point. Since the radiative effect depends on the change of dust emissions as well as the absolute dust abundance (that determine the direct radiative forcing), showing the correlation coefficient of the trends is not enough and a comparison absolute dust AOD will also be required also for the available data.

We absolutely agree that the new dust scheme is an important component of this work, that we need to demonstrate that it performs better than previous schemes, and that Fig. 1 is not sufficient for this. We've actually already done so in previous work, in which we extensively tested the new parameterization against an array of measurements and observations, including small-scale dust flux measurements (Fig. 5 in Kok et al., ACP, part 1, 2014), absolute values of AERONET AOD in dusty regions – as the reviewer suggests (Figs. 5 and S5 in Kok et al., ACP, part 2, 2014), dust optical depth and dust mass path over the tropical North Atlantic Ocean (Fig. 6 in Kok et al., ACP, part 2, 2014), and measurements of dust surface concentration and deposition (Figs. 7, 8, and S6 in Kok et al., ACP, part 2, 2014). However, we indeed do not mention this in the main text, so we've added a sentence to the "Climate model simulations for estimating the direct dust-climate feedback" to make this more clear to the reader.

Other comments:

Line 116 "most CMIP5 models are unable to reproduce historical trends in dust loading ... ", when discussing the feedback, the simulation of temperature is as important as dust, how about the performance of your CESM in reproducing the historical trends of T compared to observations and CMIP5?

That's a good question. We've added a subclause to this sentence noting that many CMIP5 models (including CESM) produce relatively good agreement against the historical temperature record. However, because the dust climate feedback is normalized by the surface temperature change, the model's ability to accurately forecast surface temperature changes is less important than it would be for determining, for example, the dust radiative forcing by 2100 (we now note this in the last paragraph of the Methodology). Also because the present article is close to the length limit, we thus think the article would not benefit from a more detailed discussion of CESM's ability to reproduce historical temperature trends relative to CMIP5.

Line 121 "evaluate whether the model can reproduce past changes in dust emissions", how did you treat the change of land use in simulations of past trends? Did you fix the land use as in the case of simulations for future?

We indeed did not make this clear – thank you for noticing that. We've now specified in the Methods that land cover is held fixed for both future and historical simulations with a plant functional type distribution that is representative of the year 2000. We did this because recent analyses indicate that changes in the West African dust plume, as measured by the AVHRR satellite, are driven almost entirely by changes in wind speed, and not land use (Kim et al., JGR, 2014; Evan et al., 2016). We therefore did not want possible errors in the description of land use changes to affect the comparison.

Fig. 1, I would like to see a comparison of the absolute AOD as well.

A quantitative comparison between the measured and modeled AOD is complicated by the fact that measurements were discontinuous at most sites, such that for instance the annually-averaged AOD resulting from averaging all available daily measurements within a given year is highly dependent on the exact days for which AOD data is available. We therefore report AOD anomalies relative to the seasonal cycle of AOD, which allows us to treat each daily measurement as a deviation from this seasonal cycle. This puts all measurements on an equal footing and eliminates (most of) the effect that the exact timing of measurements has on the resulting comparison between data and models. We've made some revisions to the supplementary text to clarify this procedure.

A quantitative comparison of the retrieved and simulated AOD anomaly for the eight stations is provided in Supplementary Figure 2, and the measured seasonal cycle for each site is shown in Supplementary Figure 8.

I also have a general question about the observations and simulations used here. How about the agreement between AVHRR and AERONET data for overlapped sites and periods of your simulations?

Since AVHRR is retrieved over the ocean (see Evan and Mukhopadhyay, 2010) and AERONET is by necessity located on land, there unfortunately is no overlap that we could use to evaluate the agreement between these two data sets.

Difference reanalysis data were used in simulations AVHRR and AERONET, how will they make a difference on the simulated absolute AOD for the overlapped period when both reanalysis data are available? Will the conclusion still hold?

That's another good question. The overlap period is from 1995-2008, for which there is no clear long-term trend in the AVHRR data, since the decline in dust emissions occurs in the 80s and early 90s. However, we can compare the model simulation results driven by MERRA and ERA-I reanalysis with the AERONET data from 1995 onward. We indeed find that our conclusion holds: CESM with the PHYS module performs substantially better against AERONET long-term trends than CESM with the BASE module. This is noted towards the bottom of the "CESM simulations of past and future climate-induced changes in the global dust cycle" subsection in the Methods section:

"However, instead using the MERRA reanalysis similarly yields a better agreement of the PHYS-AERONET simulation against the AERONET data than does the BASE-AERONET simulation."

Line 208, " $\kappa = 4.3$, ... which is consistent with the dust cycle response to climate changes simulated by the CMIP5 ensemble", I don't think a κ of 0.43 is consistent with a κ of 0.013 for CMIP5 ensemble. Please make it more accurate, e.g., within the 95% CI.

Thank you for pointing this out – this is indeed a bit unclear. The 4.3 is in units of percent, so falls within the 95% CI of the CMIP5 results, which roughly spans from -5% to +7% per unit K. We've implemented your suggestion for making this sentence clearer.

Another question is that the base module of your CESM lies near the edge of CMIPs, could you explain why you choose this model instead of a model giving results in the middle of all CMIPs. In that case, CESM with PHYS may also produce results within 95% CI.

That's a good point. Before running the simulations, we did not know where exactly the CESM results would fall, and thus were unable to choose a model that was close to the median of the CMIP5 results. However, we chose to drive the CESM model with ECHAM5 meteorology because that model was close to the median of CMIP3 models in predicting changes in desert extent (see Fig. 1 in Mahowald, 2007).

The reviewer is correct that choosing a model close to the median CMIP5 result might very well cause the PHYS results to fall within the CMIP5 95% CI. However, our main point is that the additional physical links

between climate and dust emission in the improved parameterization, which comparisons against data suggest better captures the real world (Fig. 1; Kok et al., 2014a, b), results in a greater sensitivity of the dust cycle to climate changes. In other words, implementing this parameterization in other CMIP5 models would be likely to similarly enhance the dust cycle's response, thereby suggesting that current models underestimate the dust cycle's response to climate change, as we conclude. So this conclusion is not dependent on whether the PHYS CESM result falls within or outside the CMIP5 CI.

Fig. 5, the panels b and c look like mirror images, why?

Yes, they indeed are almost opposite. That is because panel b uses the CMIP5 kappa value at the lower CI, which is -0.053 K^{-1} (see Fig. 3), whereas panel c uses the CMIP5 kappa value at the upper CI, which is $+0.073 \text{ K}^{-1}$. Since the color spacing is logarithmic, this results in nearly opposite-looking figures. To avoid confusion, we now note this explicitly in the figure's caption.

Technical corrections:

Page 1, author list, "Ward7" should be "Ward2"

Line 62, " Feedbacks in the climate system are defined as" should be " Feedbacks in the climate system are defined according to the equation"

Line 63, please use "m-2" instead of "/m2"

Line 196, the unit of kappa is missing

Thank you! We've made these corrections as suggested.

Comments from reviewer #3:

The paper entitled « Estimation of the direct dust-climate feedback » presents an original work about the direct radiative effect of dust particles on climate. The authors have used recent constraints on dust loading and global climate model simulations to constrain the direct dust-climate feedback. This study brings new elements on the impact of dust aerosols in future climate, and suggests that dust aerosols could account for a significant part of the total aerosol feedbacks while they are often neglected up to now. The use of new constraints on dust loadings also enables to reduce the uncertainties and reinforces the strengthness of the results. That is why I recommend the publication of this paper in Nature Communications, after the revision of the few following minor points.

Thank you very much for your positive and constructive comments.

- As underlined by the authors, one of the main limitation of the paper is the absence of the indirect effect in the estimation of the dust-climate feedback. The authors could add some estimations of the quantification of indirect effect as realized in previous studies, even if the uncertainties are important. It could help the reader to understand the importance of the work on the direct effect done in this study, and what remains to do as far as the indirect effect is concerned.

We appreciate this suggestion, which we indeed considered in writing this paper. However, after surveying the literature on indirect effects specifically ascribed to dust, we concluded that (i) very few such studies exist (DeMott et al., 2010; Sagoo and Storelvmo, 2017), and (ii) even the qualitative model results diverge (e.g., more dust produces cooling in Sagoo and Storelvmo (2017), but warming in DeMott et al. (2010)). The physics underlying these different responses is complex, and also differs substantially for mixed-phase versus cirrus clouds. Therefore, an accurate discussion of the dust indirect climate feedback would require substantial effort that would be going beyond the scope of this paper (especially considering that the present paper is already close to the length limit). We therefore decided to instead possibly revisit this issue in a future paper.

However, in response to both this comment and a related comment from referee #1, we now explicitly point out in the discussion that indirect dust climate feedbacks could potentially substantially exceed the direct dust climate feedback.

- I have some difficulties to understand what the authors call the « regional dust-climate feedback », and what is really behind this estimation, as they do not consider spatial variations in the local change in dust loading per degree (lines 99-100). The authors should clarify this point.

This is indeed a bit confusing. The spatial variability in the regional dust climate feedback can be driven by variability in both the DRE and the dust loading response, κ (see Eq. 5). However, we cannot reasonably constrain κ on a regional basis with the current generation of CMIP simulations (see response to previous comment by referee #1), such that we do not include spatial variability in κ and instead assume that the regional dust changes equal the global dust change (i.e., that $\kappa(\theta, \phi) = \kappa$). Therefore, the spatial variability in the direct dust-climate feedback derives from variability in the present-climate DRE, which we can reasonably constrain (see Methods and Supplementary Figures 5-7).

To avoid this confusion for readers, we've now noted this explicitly in the sentence formerly on line 99-100.

- CMIP5 simulations: the authors should give details on the selection of models. Do they consider all the CMIP5 simulations which are available, or have they established a subset of models ? (line 108)

We use all CMIP5 models that had prognostic dust, following Allen et al. (2016). We now note this explicitly in that sentence.

- Table1: is it a problem to use meteorology from a CMIP3 simulation in the future CEMS simulations compared to CMIP5 simulations for the other models ? The authors should at least give the scenario which has been used for this CMIP3 simulation.

It indeed might have been preferable for consistency to use CMIP5 meteorology to drive the CESM simulations. However, this is what was available to us at the time (the simulations were performed a few years ago) and, considering the many uncertainties, we do not think that the model changes between the CMIP3 and CMIP5 model versions are a major contributor to the uncertainty.

The scenario used for the ECHAM5 CMIP3 simulations (A1B) is noted in the subsection "CESM Simulations of the response of dust emissions to future climate changes" in the Methods section.

- Have the authors considered the use of different scenarios for this study ? Would the results differ with scenarios with a less important global warming ?

That is a good question. One of the advantages of the feedback parameter is that it is normalized with respect to the temperature, which inherently makes it substantially less sensitive to the exact scenario used by a climate model than, say, the dust radiative forcing. That being said, we did not run different scenarios so we cannot quantify the sensitivity of the results to the exact scenario.

To address this comment, we've added two sentences acknowledging this limitation in the last paragraph of the Methods section.

- Land use change (line 110): I am not sure that all CMIP5 simulations include land use change, so maybe there is a possibility to separate two kinds of simulations ? Moreover have the others carried out also CESM simulations with land use, in order to check the impact of land use on the evolution of dust loadings ?

All the used CMIP5 simulations follow the RCP8.5 scenario, which includes land use changes as specified in the cited Moss et al. (2010) and Hurtt et al. (Climatic Change, 2011; not explicitly cited due to citation limit). And indeed, the cited Ward et al. (2014) used CESM simulations to attribute a fraction of dust loading changes to land use changes, which we discuss in the first paragraph of "Estimation of the direct dust-climate feedback."

- Figure 1: I am surprised on the large negative bias in dust AOD in the simulation using BASE module compared to the one using PHYS module. Is there a specific tuning only in PHYS module ?

Both simulations were tuned to minimize the disagreement with AERONET AOD in dusty regions, per the procedure specified in Kok et al. (2014, part 2, p. 13,048), and summarized in the last sentence of "CESM simulations of past and future climate-induced changes in the global dust cycle" in the Methods section. The reason for the large bias is that the BASE model emits too much dust in the eastern part of North Africa, and not enough in the western part (see Figs. 4 and 5 in Kok et al., 2014, part 2). This seems to be a common problem for models: the analysis of Evan et al. (2014) indicates that most CMIP5 models underestimate the dust AOD over the tropical North Atlantic Ocean. The new parameterization corrected this problem, and thus also increased the dust AOD in the tropical North Atlantic, in better agreement with the AVHRR results.

- Lines 173-176: How is it possible to have both a « slight increase in wind speed over most North African dust source regions » and « a decrease in dust emissions due to North African wind speed changes » ? Is it instead an increase in dust emissions, as we can see in Fig. S3 ?

The wording here is indeed a bit confusing. We mean to say that some of the CMIP5 models also find this slight increase in wind speed, although the mean of all CMIP5 simulations predicts a slight decrease. We've revised this sentence to communicate this more clearly.

- Figure 3: I wonder if the authors have tried to understand the distribution of the different CMIP5 simulations. Is there any link between the models with a strong warming, and the models with a strong increase in dust loading ?

That's a good question. We've included plots for the CMIP5 predictions of both dust loading change and kappa as a function of surface T change. There are no clear trends apparent, in part because the model spread is so large (as also expressed in Figs. 3 – 5). We thus do not discuss these results explicitly in the paper, as they do little inform the main objective of the paper, to estimate the direct dust climate feedback.

- Figure 5: Following my remark on regional dust-climate feedback, I am also surprised by the important difference between Fig 5a and Fig 5b, as the sign of the feedback has changed. The authors should explain more in details how you can have such changes in the sign of the feedback.

This is indeed a bit confusing and deserving of additional explanation (also see response to similar comment from reviewer #2). We've therefore added the following sentence to the caption of Fig. 5: "Since the probability distribution of κ derived from the CMIP5 simulations spans both negative and positive values, the sign of the dust climate feedback in panel (b) is opposite that of the other panels (see Fig. 3)."

Reviewers' comments:

Reviewer #1 (Remarks to the Author):

I am satisfied with the reviewers' responses to my questions and comments, and find the revised paper suitable for publication in Nature Communications.

Reviewer #2 (Remarks to the Author):

I have to say that I was a bit upset when the authors refused to provide the requested information by arguing about the length limit "Also because the present article is close to the length limit, we thus think the article would not benefit from a more detailed discussion...". The authors could simply show it in the response which is a common practice of scientific communications.

Previous iterations:

"

Line 116 "most CMIP5 models are unable to reproduce historical trends in dust loading ... ", when discussing the feedback, the simulation of temperature is as important as dust, how about the performance of your CESM in reproducing the historical trends of T compared to observations and CMIP5?

That's a good question. We've added a subclause to this sentence noting that many CMIP5 models (including CESM) produce relatively good agreement against the historical temperature record. However, because the dust climate feedback is normalized by the surface temperature change, the model's ability to accurately forecast surface temperature changes is less important than it would be for determining, for example, the dust radiative forcing by 2100 (we now note this in the last paragraph of the Methodology). Also because the present article is close to the length limit, we thus think the article would not benefit from a more detailed discussion of CESM's ability to reproduce historical temperature trends relative to CMIP5.

"

Comments:

Just to make it more clear. My question is not about the performance of previous CMIP5. My concern is that, after implementing the new dust scheme, will the modeled temperature be better (or worse) than that with the old dust scheme? I asked this because this paper is about the temperature/climate-dust feedback. If your model produced good results for both dust emission and temperature, people will be confident about the new schemes and trust the modeled dust-climate feedback. If your model produced good dust emission but bad temperature, it will lead to doubt your simulated dust-climate feedback. That's why I asked for the performance of your CESM in reproducing historical trends of T compared to observations.

Previous iterations:

"

Fig. 1, I would like to see a comparison of the absolute AOD as well.

A quantitative comparison between the measured and modeled AOD is complicated by the fact that measurements were discontinuous at most sites, such that for instance the annually-averaged AOD resulting from averaging all available daily measurements within a given year is highly dependent on the exact days for which AOD data is available. We therefore report AOD anomalies relative to the seasonal cycle of AOD, which allows us to treat each daily measurement as a deviation from this seasonal cycle. This puts all measurements on an equal footing and eliminates (most of) the effect that the exact timing of measurements has on the resulting comparison

between data and models. We've made some revisions to the supplementary text to clarify this procedure.

"

Comments:

The authors argued that they could not make a comparison of the absolute AOD because the annually-averaged AOD is highly dependent on the number of days with data that is available. I don't understand this argument. The anomaly is defined as a departure from a reference value or average. If the averaged values are problematic, how can you trust the anomaly? Or the AOD comparison is not good?

Reviewer #3 (Remarks to the Author):

The authors have addressed all the points that I have mentioned in my review, and have brought new elements to clarify these different points. The article has been modified and is now ready to be published.

We thank reviewer #2 and the editor for their additional comments. Below we include a point-by-point response to the comments, and describe the corresponding changes we have made to the manuscript. We hope these responses and revisions have addressed all comments satisfactorily. All responses below are in blue text.

Reviewer #2 (Remarks to the Author):

I have to say that I was a bit upset when the authors refused to provide the requested information by arguing about the length limit "Also because the present article is close to the length limit, we thus think the article would not benefit from a more detailed discussion...". The authors could simply show it in the response which is a common practice of scientific communications.

We very much regret that our response was not satisfactory. We indeed misinterpreted your comment on the temperature comparison (thank you for clarifying that – we've included a detailed response below), and you were entirely correct about the AERONET comparison. It of course is possible to include a meaningful comparison of the annually-averaged aerosol optical depth between AERONET observations and the simulations with the old and the new dust schemes. I apologize for not thinking this through carefully enough in our first revision, and thank you for pointing that out.

Just to make it more clear. My question is not about the performance of previous CMIP5. My concern is that, after implementing the new dust scheme, will the modeled temperature be better (or worse) than that with the old dust scheme? I asked this because this paper is about the temperature/climate-dust feedback. If your model produced good results for both dust emission and temperature, people will be confident about the new schemes and trust the modeled dust-climate feedback. If your model produced good dust emission but bad temperature, it will lead to doubt your simulated dust-climate feedback. That's why I asked for the performance of your CESM in reproducing historical trends of T compared to observations.

Thank you for clarifying your comment – we indeed misunderstood your original suggestion.

This is a good question. We have not investigated whether CESM with the new dust scheme reproduces the observed observational record any better than with the old scheme. We have not done so because we unfortunately consider such an investigation not informative to the accuracy of the constraints on the dust-climate feedback that our paper presents. There are a few reasons for this:

- First, using a comparison against the observed temperature record to test the fidelity of the simulated dust cycle and its radiative impact is highly unusual; indeed, we are not aware of any study that has used agreement against the historical temperature record to inform the accuracy of the model's dust cycle or its radiative effects (e.g., Ginoux et al., 2001; Woodward, 2001; Zender et al., 2003; Miller et al., 2006; Zhao et al., 2010; Huneeus et al., 2011; Stanelle et al., 2014). A primary reason for this is that the agreement of any climate model against the observed temperature record is dependent on a host of factors that are independent of the model's ability to capture the global dust cycle. Furthermore, models are generally tuned to reproduce the observational record (e.g., Flato et al., 2013), as implied by the inverse relation between the strength of aerosol cooling and climate sensitivity (Fig. 1 in Kiehl (2007)). The consequent presence of canceling errors in climate models means that an improvement in the model's ability to represent one of the many components that affects the observed temperature record, in this case the dust cycle, does not necessarily produce an improvement in the model's ability to capture the observational record. Therefore, we would be unable to draw conclusions from a model comparison against the observed temperature record.
- Second, it is unclear whether dust cools or warms on a global basis (Fig. 4 in Kok et al. (2017)), or even on a regional basis (see Supplementary Figs. S5). Consequently, the sign of the dust radiative effects simulated by any particular model is determined by the specific optical properties assumed and radiative transfer model used. Since CESM produces a relatively strong dust warming relative to other models (see Supplementary Fig. S5 in Kok et al. (2017)), we would not be able to draw meaningful conclusions from an investigation of whether or not CESM with the new dust scheme reproduces the observed temperature record any better. This is also why we use an ensemble of (bias-corrected) climate models to constrain the regional dust radiative effect and the resulting climate feedback (see Kok et al. (2017) and Supplementary Fig. S5).

- Third, investigating whether the temperature record is reproduced more accurately with the new dust scheme requires the dust radiative effects to be coupled to the atmospheric dynamics. This requires the model to simulate atmospheric fields, rather than use reanalysis meteorology, as done in the reported simulations. Such an approach would be problematic, as previous work (Ridley et al., 2014; Evan et al. 2014, 2016) has shown that model errors, most likely in simulated wind fields, prevent models from capturing the observed changes in North African dust emissions over the past few decades. In other words, capturing observed changes in North African emissions is currently only possible when models are forced with reanalysis meteorology. This also has important implications for the accuracy with which models can simulate the direct dust-climate feedback. The manuscript discusses this in detail at the end of the Methods section and in the discussion Section. Such a simulation would thus not be useful, as it is unlikely to capture dust emission changes over North Africa for reasons that are independent of the dust emission scheme. Making such a simulation useful would require improvements in model predictions of atmospheric dynamics, and particularly wind speeds over Northern Africa, which is clearly beyond the scope of this paper.
- The fourth and perhaps most straightforward reason why we consider a detailed comparison of CESM's simulated temperature record against observations to not be worthwhile is that the assessment of the dust-climate feedback in this paper is entirely independent of CESM's ability to accurately reproduce past and forecast future temperature changes. That is, we force CLM (CESM's land model) with output from the middle-of-the-road climate model ECHAM5, not with CESM's atmospheric model (see Methods for details and reasoning). Furthermore, none of the CMIP5 simulations were done with CESM. As such, whether or not CESM with the improved dust scheme shows an improvement in its ability to reproduce observed temperature trends has no bearing on the accuracy of our constraints on the dust-climate feedback.

For these reasons, the requested additional work would not affect the constraints on the dust-climate feedback that our paper poses (Figs. 3-5), or provide information beyond what we already discuss in the last few sentences of the Methods section on how accurate those constraints might be. In addition, performing the proposed analysis would require us to run a separate set of simulations in which dust is radiatively coupled to the rest of the climate system, which requires substantial additional time and computational resources. We therefore respectfully request that the revised version be considered for publication without the proposed addition.

The authors argued that they could not make a comparison of the absolute AOD because the annually-averaged AOD is highly dependent on the number of days with data that is available. I don't understand this argument. The anomaly is defined as a departure from a reference value or average. If the averaged values are problematic, how can you trust the anomaly? Or the AOD comparison is not good?

The reviewer is entirely correct on this point: it is possible to include a meaningful comparison of the annually-averaged aerosol optical depth between AERONET observations and the simulations with the old and the new dust schemes. We apologize for not thinking this through carefully enough in our first revision. We have now included this in the revised paper and updated the supplementary text accordingly. This revised figure shows more clearly what previous results (Supplementary Fig. 2 in the previous version and Fig. 5 in Kok et al. (2014)) already showed: that the new dust model better reproduces both the magnitude and the long-term trend of AOD in dusty regions.

References

- Flato, G., et al., 2013. Evaluation of Climate Models, in: Stocker, T.F., Qin, D., Plattner, G.-K., Tignor, M., Allen, S.K., Boschung, J., Nauels, A., Xia, Y., Bex, V., Midgley, P.M. (Eds.), *Climate Change 2013: The Physical Science Basis. Contribution of Working Group I to the Fifth Assessment Report of the Intergovernmental Panel on Climate Change*. Cambridge University Press, Cambridge, United Kingdom and New York, NY, USA.
- GINOUX, P., CHIN, M., TEGEN, I., PROSPERO, J.M., HOLBEN, B., DUBOVIK, O., LIN, S.J., 2001. Sources and distributions of dust aerosols simulated with the GOCART model. *J. Geophys. Res.* 106, 20255-20273.

- Huneus, N., et al., 2011. Global dust model intercomparison in AeroCom phase I. *Atmos. Chem. Phys.* 11, 7781-7816.
- Kiehl, J.T., 2007. Twentieth century climate model response and climate sensitivity. *Geophysical Research Letters* 34.
- Kok, J.F., Albani, S., Mahowald, N.M., Ward, D.S., 2014. An improved dust emission model - Part 2: Evaluation in the Community Earth System Model, with implications for the use of dust source functions. *Atmos. Chem. Phys.* 14, 13043-13061.
- Kok, J.F., et al., 2017. Smaller desert dust cooling effect estimated from analysis of dust size and abundance. *Nature Geoscience* 10, 274-278.
- Miller, R.L., et al., 2006. Mineral dust aerosols in the NASA goddard institute for Space Sciences ModelE atmospheric general circulation model. *J. Geophys. Res.-Atmos.* 111, D06208.
- Stanelle, T., Bey, I., Raddatz, T., Reick, C., Tegen, I., 2014. Anthropogenically induced changes in twentieth century mineral dust burden and the associated impact on radiative forcing. *Journal of Geophysical Research-Atmospheres* 119, 13526-13546.
- Woodward, S., 2001. Modeling the atmospheric life cycle and radiative impact of mineral dust in the Hadley Centre climate model. *Journal of Geophysical Research-Atmospheres* 106, 18155-18166.
- Zender, C.S., Bian, H.S., Newman, D., 2003. Mineral Dust Entrainment and Deposition (DEAD) model: Description and 1990s dust climatology. *Journal of Geophysical Research-Atmospheres* 108, 4416.
- Zhao, C., et al., 2010. The spatial distribution of mineral dust and its shortwave radiative forcing over North Africa: modeling sensitivities to dust emissions and aerosol size treatments. *Atmospheric Chemistry and Physics* 10, 8821-8838.

Reviewers' Comments:

Reviewer #2:

Remarks to the Author:

I still have one question:

The new AOD plots (Fig. S1) show that at Dakar and Cape Verde sites, the PHYS module substantially overestimates the AOD while the BASE module shows good agreements. This result seems to be contradictory to Fig 1, where the BASE dust module substantially underestimates the AOD off the West-African coast and the PHYS module helps resolve these problems.

Any explanation?

We thank reviewer #2 for his/her careful reviews, and address his/remaining comment below.

Reviewer #2 (Remarks to the Author):

I still have one question:

The new AOD plots (Fig. S1) show that at Dakar and Cape Verde sites, the PHYS module substantially overestimates the AOD while the BASE module shows good agreements. This result seems to be contradictory to Fig 1, where the BASE dust module substantially underestimates the AOD off the West-African coast and the PHYS module helps resolve these problems.

Any explanation?

That is a good question. There are several possible reasons that could contribute to this result. First, the AERONET stations are point measurements, whereas the West-African AOD is averaged over a large area, making it less sensitive to model errors in the exact spatial distribution of the dust. Second, the comparisons against AERONET and the West-African dust AOD are from two different simulations (BASE/PHYS-AVHRR and BASE/PHYS-AERONET), which are driven by different reanalysis meteorology data sets (the rationale for this is explained in Methods). Differences in meteorology, especially wind speed, could cause the overestimation at Cape Verde and Dakar for BASE/PHYS-AERONET while reproducing the correct magnitude of dust AOD off of the West-African coast.